# Melting conditions in the modern Tibetan crust since the Miocene

Jinyu Chen [1,2], Fabrice Gaillard[2], Arnaud Villaros[2], Xiaosong Yang[1], Mickael Laumonier[3], Laurent Jolivet[2,4], Martyn Unsworth[5], Leïla Hashim[6], Bruno Scaillet [2] & Guillaume Richard[2]

Abundant granitic rocks exposed in ancient mountain belts suggest that crustal melting plays a major role in orogenic processes. However, complex field relations and superposition of multiple tectonic events make it difficult to determine the role of melting in orogenesis. In contrast, geophysical measurements image present-day crustal conditions but cannot discriminate between partial melt and aqueous fluids. Here we connect pressure–temperature paths of Himalayan Miocene crustal rocks to the present-day conditions beneath the Tibetan plateau imaged with geophysical data. We use measurements of electrical conductivity to show that 4–16% water-rich melt is required to explain the crustal conductivity in the northwestern Himalaya. In southern Tibet, higher melt fractions >30% reflect a crust that is either fluid-enriched (+1% $H_2O$) or hotter (+100 °C) compared to the Miocene crust. These melt fractions are high enough for the partially molten rocks to be significantly weaker than the solid crust.

[1] State Key Laboratory of Earthquake Dynamics, Institute of Geology, China Earthquake Administration, 100029 Beijing, China. [2] Université d'Orléans, CNRS, BRGM, ISTO, UMR 7327, F -45071 Orléans, France. [3] Laboratoire Magmas et Volcans, Campus Universitaire des Cézeaux, 6 Avenue Blaise Pascal, 63178 Aubière Cédex, France. [4] Sorbonne Université, CNRS-INSU, Institut des Sciences de la Terre Paris, ISTeP, UMR 7193, F-75005 Paris, France. [5] Department of Earth and Atmospheric Sciences, University of Alberta, Edmonton, AL T6G 2J1, Canada. [6] Department of Earth Science, University of Minnesota – Twin Cities, 55455Minneapolis, MN, USA. Correspondence and requests for materials should be addressed to J.C. (email: jinyu@ies.ac.cn) or to F.G. (email: fabrice.gaillard@cnrs-orleans.fr)

Crustal melting produces granitic liquids and is a fundamental process by which the continents have differentiated into a depleted and dry lower layer and an upper enriched crust[1]. During continental collision, a combination of processes and properties such as crustal thickening, exhumation, radioactive heating, shear heating, and the temperature dependence of thermal diffusivities yield temperatures high enough (>750 °C) to trigger significant melting events in the crust[2–5]. Crustal melting dominantly results from the presence of water in the rocks as dry lithologies would otherwise require much higher temperatures. Petrologists have distinguished two cases of hydrous crustal melting: fluid-absent melting[2], also called dehydration melting, where water exclusively results from breakdown of water-bearing minerals such as micas at temperature of ≥750–850 °C, or fluid-present melting, where in addition to water-bearing minerals, a free fluid phase ("free water") embedded in the porosity at subsolidus conditions enables melting at lower temperatures (<750 °C). Both cases produce water-rich melts (with >5 wt% $H_2O$), but fluid-present melting broadly tends to produce more abundant melts that are richer in water[6]. The presence of a widespread free fluid phase in the deep crust and its link with crustal melting remains a strongly debated issue, including in the Himalayas[1,6–8].

Decompression associated with exhumation marking the late stages of orogens is often associated with episodes of crustal melting with voluminous granitic rocks and migmatites[1]. In addition, melting events are being increasingly reported through all stages of orogenesis[9–11]. At the orogen scale, the presence of melt can trigger the weakening of vast crustal regions[12,13] in which large deformations can be accommodated[9]. Understanding how melting events occur (both in magnitude and rate) and the role they play in orogeny represents a fundamental question in geodynamics.

Evidence of past presence of melting events can be observed within anatectic granites and migmatites that record multiple stages of melting, migration, and solidification, sometimes resolved in ages and timing at a local scale[14,15]. Geophysical imaging allows large-scale and present-day conditions of active orogens to be determined (e.g., Nelson et al.[16]; Unsworth et al.[17]) and can be used as a constraint on geodynamic models (e.g., Jamieson et al.[18]). However, the link between geophysical observations and petrological processes suffers from non-uniqueness. This is highlighted with the unresolved interpretation of the various Himalayan geophysical bright spots[16,19–22], where it has proven difficult to distinguish between aqueous fluids and melt, since both have high electrical conductivity and low seismic velocity signatures. In order to address this issue, we hereafter briefly present the geological structure of the Himalayan chain and summarize the geophysical data so far obtain at crustal scale.

The Himalayan orogen is >2000 km long and is located between the Indus-Tsangpo suture zone (ITS) to the north and the Main Frontal Thrust (MFT) to the south. It was formed by the Cenozoic collision of India and Asia, which has accommodated at least 1400 km of crustal shortening resulting in crustal thickening, metamorphism, and partial melting[5,17,23–25] (Fig. 1a). From north to south, four parallel, laterally continuous tectonostratigraphic units compose the orogenic wedge (Fig. 1b, c): the Tethyan Himalayan Sequence (THS), the Greater Himalayan Sequence (GHS), the Lesser Himalayan Sequence (LHS), and the Sub-Himalayan Sequence (SHS)[22,23,25]. The THS (1840–40 Ma) consists of Proterozoic to Eocene siliciclastic and carbonate sedimentary rocks interbedded with Paleozoic and Mesozoic volcanic rocks, while the main lithology of LHS (1870–850 Ma) includes metasedimentary rocks, metavolcanic rocks, and gneiss[25]. The GHS (~1800–480 Ma), composed primarily of high-grade gneisses, metapelites, and pervasive migmatites, is commonly interpreted to be the result of the southward extrusion, over a distance of >100 km, of a low-viscosity mid-crustal channel sandwiched between two major, broadly coeval, shear zones, i.e., the Main Central Thrust (MCT) at the base and the South Tibetan Detachment (STD) at the top[5,24,25]. This channel flow model is, however, disputed by studies indicating instead alternating periods of shortening and extension in the Himalaya[26,27]. Both MCT and STD were active >13–11 Ma ago and their cessations are seemingly coeval[28]. The STD stopped moving when the thrust front jumped southward to the MBT and the MFT[28] (Fig. 1). A significant part of exhumation was accommodated by tectonic unroofing thanks to the normal sense displacement along the STD before 11 Ma and probably more by erosion since that time. The High Himalayan Leucogranites (HHL) are plutonic rocks emplaced 26–11 Ma ago at ca. 10–12 km depth in the GHS; the MCT and STD were mostly operating when these granites were solidified[27]. The Tethyan Himalayan leucogranites (THL) were emplaced in the THS geological unit 28–7 Ma ago[29]. HHL and THL leucogranitic melts were extracted from the GHS[29,30] at a depth of >20 km[29–31] during the late Oligocene–Miocene. In addition, the involvement of variable amounts of fluids derived from the underlying LHS unit and affecting melting of the GHS has been suggested from geochemical studies[29], though most HHL are believed to be derived from fluid-absent melting[24,32,33]. So far, whether the continuous processes in the tectonic regime affecting the primary Himalayan faults implies a cessation of melting beneath Tibet or not constitutes a major issue for our understanding of the mechanical behavior of the crust beneath the Tibetan plateau[5,23,24,32].

Geophysical data contribute to this debate by providing present-day images of the crust that can potentially indicate the presence of crustal melts. Magnetotelluric (MT) data show several high conductivity regions currently located in the mid-crust of the Tibet–Himalaya orogen[16,17,19,20,34,35] that are also characterized by low seismic velocities[36–39] (Supplementary Table 1).

In the north-western Himalaya, the electrical conductivity of these layers ranges from ~0.03 S/m at 15–20 km depth south of the ITS to 0.1–0.2 S/m at 20–25 km depth north of the ITS and 0.05–0.1 S/m at 10–13 km depth beneath the ITS[34] (positions P1, P2, and P4 in Fig. 1b, respectively). We also note the recent MT work in north-western Himalaya reporting high electrical conductivity layers south of the MCT[40], but these are clearly unrelated to melting due to the very cold crust. In the southern Tibet, higher conductivities have been reported with values of ~0.3 S/m and 0.05–0.1 S/m at a depth of 18–20 km and 10–13 km, respectively[17] (positions P3 and P5 in Fig. 1c, respectively). These conductivity anomalies, localized with respect to the original MT work by Unsworth et al.[17] and Arora et al.[34] in Supplementary Fig. 1, may reveal melting in the Tibetan crust, but determining the presence of crustal melts from conductivity–depth relationships is not straightforward as critical experimental constraints are missing and the geological solution to high crustal conductivities may not be unique.

Here we address this issue with a new approach that combines experimental conductivity measurements of melts and pressure–temperature paths of ancient exhumed metamorphic rocks, so that an electrical conductivity–depth relationship can be derived and compared with the present geophysical observations. It will be shown that this greatly reduces the non-uniqueness in interpreting high conductivity regions and also gives consistency with the geophysical signal measured beneath the Tibetan plateau. This study concludes that a large fraction of melt (16–35%) must operate beneath the Tibetan plateau, sufficient to significantly weaken crustal rocks. A similar magmatic process has been operating in the Himalaya for the past 20 Ma.

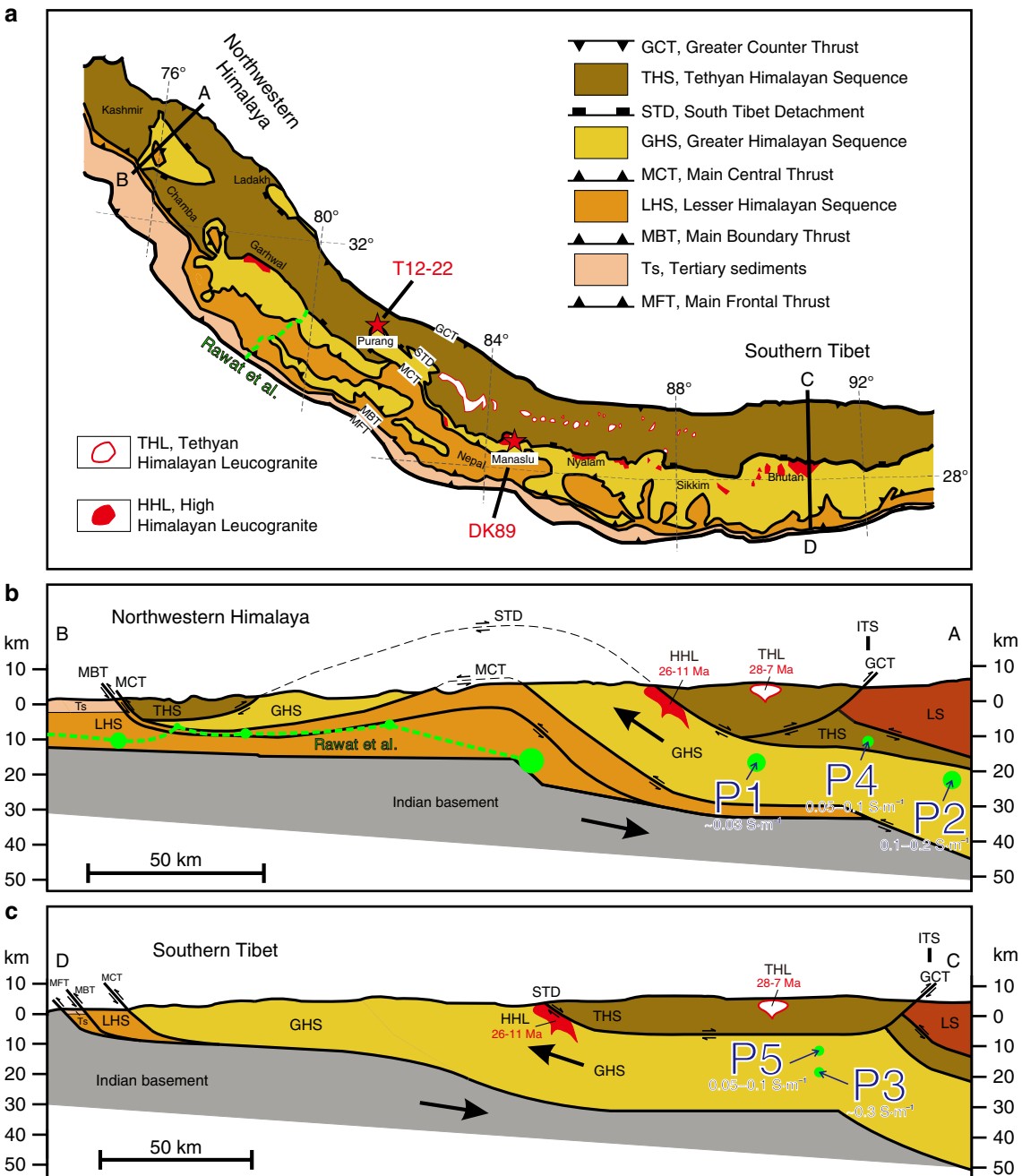

**Fig. 1** Simplified geological map combined with the cross-sections of the northwestern Himalaya and southern Tibet. **a** The leucogranites used for electrical conductivity measurements (T12-22 in Purang county and DK89 in the base of the Manaslu pluton[41], red stars on the geological map) were collected in the GHS near the STD. The location of the MT profile probed by Rawat et al.[40] is labeled by the green dashed line. **b** In the northwestern Himalaya (cross-section A, B), the high electrical conductivity spots of ~0.03 S/m (15–20 km), 0.1–0.2 S/m (20–25 km), and 0.05–0.1 S/m (10–13 km) identified by magnetotelluric data (MT)[34] are labeled P1, P2, and P4, respectively. The green dashed line and circles, respectively, identify the high electrical conductivity layer and spots, as probed by Rawat et al.[40]. **c** The other spots at ~0.3 S/m (18–20 km) and 0.05–0.1 S/m (10–13 km)[17] are labeled P3 and P5 in the southern Tibet (cross-section C, D). The leucogranites emplaced just beneath the STD have been dated from 26 to 11 Ma, while those emplaced within the THS have been dated from 28 to 7 Ma[29]. **a**–**c** were adapted from Fig. 1 of Hashim et al.[22]. TS tertiary sediments, LHS Lesser Himalayan Sequence, GHS Greater Himalayan Sequence, THS Tethyan Himalayan Sequence, LS Lhasa terrane, ITS Indus-Tsangpo suture, MFT Main Frontal Thrust, MBT Main Boundary Thrust, MCT Main Central Thrust, STD South Tibet Detachment, GCT Great Counter Thrust, THL Tethyan Himalayan Leucogranite, HHL High Himalayan Leucogranite

## Results

**Effects of $H_2O$, $T$ and $P$ on the conductivity of crustal melts.** We measured the electrical conductivities of hydrous leucogranitic melts at high pressure (0.5–2 GPa) and temperature (750–1400 °C). We first produced hydrous granitic melts by annealing granite powders with 0–8 wt.% water at 0.4 GPa and 1000 °C (see Methods in details) and subsequently measured the electrical conductivity of the recovered hydrated samples at variable pressure–temperature conditions. The starting materials were HHL rocks (T12-22 and DK89[41]) presently outcropping in GHS near STD (Fig. 1a). T12-22 is a tourmaline-muscovite leucogranite and DK89 is a two-mica leucogranite[41]. Their chemical

**Table 1 Chemical compositions (given in wt%) of the leucogranitic rocks (T12-22a and DK89a)**

| Samples | $SiO_2$ | $TiO_2$ | $Al_2O_3$ | FeO | MgO | CaO | $Na_2O$ | $K_2O$ | $P_2O_5$ | Total |
|---|---|---|---|---|---|---|---|---|---|---|
| T12-22a | 74.15 | 0.05 | 14.73 | 0.95[a] | 0.17 | 0.74 | 4.46 | 3.95 | 0.13 | 99.40 |
| DK89a | 73.04 | 0.13 | 15.32 | 0.91[a] | 0.20 | 0.85 | 3.85 | 4.96 | 0.14 | 99.55 |

T12-22a was determined by XRF, while DK89a[41] was determined by EMPA
[a]FeO as total Fe ($FeO+Fe_2O_3$).

**Table 2 Experimental details and results of the melt conductivity measurements**

| Samples | $P$ (GPa) | $T$ (°C) | $H_2O$ (wt%) content before experiments (points) | $H_2O$ (wt%) content after experiments (points) | $Log_{10}\,\sigma_0$ (S/m) | $\Delta H^a$ (kJ/mol) |
|---|---|---|---|---|---|---|
| *T12-22* | | | | | | |
| T0(11)-0.5 | 0.36–0.60 | 777–1200 | nd | 0.05–0.08 (10) | 2.7 | 64.5 |
| T0(8)-1 | 0.90–1.37 | 466–1203 | nd | nd | 2.8 | 78.0 |
| T0(8)-2 | 1.77–2.35 | 803–1206 | nd | nd | 2.4 | 83.5 |
| T0(10)-2 | 1.81–2.47 | 1092–1305 | nd | 0.40–0.60 (10) | 2.5 | 79.8 |
| T3(9)-2 | 1.86–2.46 | 798–1100 | 2.73–4.44 (10) | 1.26–4.63 (8) | 2.1 | 57.0 |
| T6(9)-2 | 1.90–2.52 | 785–1105 | 5.01–7.50 (10) | 4.56–7.07 (10) | 2.6 | 56.1 |
| T7(1)-2 | 1.90–2.47 | 760–1018 | 6.15–8.97 (10) | 5.45–8.10 (7) | 2.4 | 50.6 |
| T7(4)-1 | 1.10–1.36 | 785–1101 | 6.29–9.12 (10) | 4.78–8.42 (10) | 2.3 | 41.5 |
| *DK89* | | | | | | |
| DK0a | 0.94–1.14 | 745–1408 | nd | nd | 2.9 | 83.6 |
| DK7a | 0.98–1.20 | 909–1407 | 5.77–8.29 (2) | 0.93–2.61 (5) | 2.4 | 62.5 |
| DK11b | 0.96–1.18 | 791–1403 | 7.83–11.97(19) | 1.05–1.71(9) | 2.6 | 53.1 |

Small variations in pressure occurred during the temperature cycles. Melt water concentration was obtained from FTIR spectroscopy and is followed by the number of measurements (in parentheses). Water concentrations in some nominally dry samples were not determined (nd) before and after experiments
nd not determined
[a]The activation enthalpy ($\Delta H$) includes the activation energy ($E_a$) and the activation volume ($\Delta V$) as follows: $\Delta H = E_a + P \cdot \Delta V$

composition is characteristic of peraluminous silicic rocks (Table 1), representative of the typical products of crustal melting in the Himalayan orogen[2,41].

Ten in situ electrical conductivity experiments (seven samples of T12-22, three samples of DK89) were conducted at three different pressures (~0.5, 1, and 2 GPa) up to 1400 °C on samples with various water contents (0–8 wt%, Table 2) (see Methods in details). Half-inch (0.5 GPa) and 3/4-inch (1 and 2 GPa) assemblages were used in end-loaded piston-cylinder apparatus at Institut des Sciences de la Terre d'Orléans (ISTO)[42,43] (Fig. 2).

After the experiments, the samples were recovered for textural and chemical observations, such as the geometry and the absence of crystals (as determined by scanning electron microscope) (Fig.2), determination of chemical composition (electron microprobe analysis and X-ray fluorescence) (Table 1 and Supplementary Table 2), and water content (Fourier transform infrared spectroscopy (FTIR)) (Table 2).

Electrical conductivity results reveal that the melt conductivity increases by one-log unit when the water content is increased to 6 wt% $H_2O$ (Fig. 3b). Its effect is almost comparable to the effect of pressure (Fig. 3a) and much stronger than that of temperature (at crustal magmatic conditions).

The conductivity results were fitted using an Arrhenius equation having the following form:

$$\sigma = \sigma_0 \cdot \exp\left(-\frac{E_a + P \cdot \Delta V}{R \cdot T}\right) \quad (1)$$

where $\sigma$ is the leucogranitic melt electrical conductivity (S/m), $\sigma_0$ is the pre-exponential factor (S/m), $E_a$ is the activation energy (J/mol), $P$ is the pressure (GPa), $\Delta V$ is the activation volume ($mm^3$/mol), $R$ is the gas constant (8.314 J/mol/K), and $T$ is the temperature (K). Because of the relatively fast heating (~200 °C/min), only the data

from the cooling stage was considered for our fit, except for sample T0(8) for which two heating–cooling cycles provided data to fit the Arrhenius relationship (Supplementary Fig. 4). The summary of experimental conditions, the regressed activation enthalpy ($\Delta H$), and pre-exponential factors ($\sigma_0$) for each individual run are available in Table 2.

We have developed an empirical model defining the $P$–$T$–$H_2O$ dependencies of $\sigma$ (S/m) over the range of water concentration ($w$ in wt%), temperature ($T$ in K), and pressure ($P$ in GPa) of this study (Fig. 3). It was parameterized as:

$$\ln\sigma_0 = 6.673 - 0.491P \quad (2)$$

$$E_a = 58987 - 2200\ln(w+0.046) \quad (3)$$

$$\Delta V = 10927 - 1981w \quad (4)$$

The average difference between calculated and measured conductivity values is equal to 0.04 log units, which is very close to the experimental uncertainties (i.e., 0.03 log units).

**Multiple interpretations of a highly conductive crust.** The cause of high conductivity in the crust has been a long-standing and puzzling question[44,45]. If aqueous fluids are involved, the conductivity should increase with fluid/rock ratio, temperature, and NaCl concentration in the fluid[19,46]. In the Tibetan setting, a model with highly saline fluids (i.e., a few percent of fluids with ~20 wt% NaCl[46]) is a possible solution, but it is challenged by the rare geological description of saline aqueous fluids in the middle Tibetan crust[22]. While numerous hot springs in Tibet may be the surficial expression of upper crustal fluids mixing with

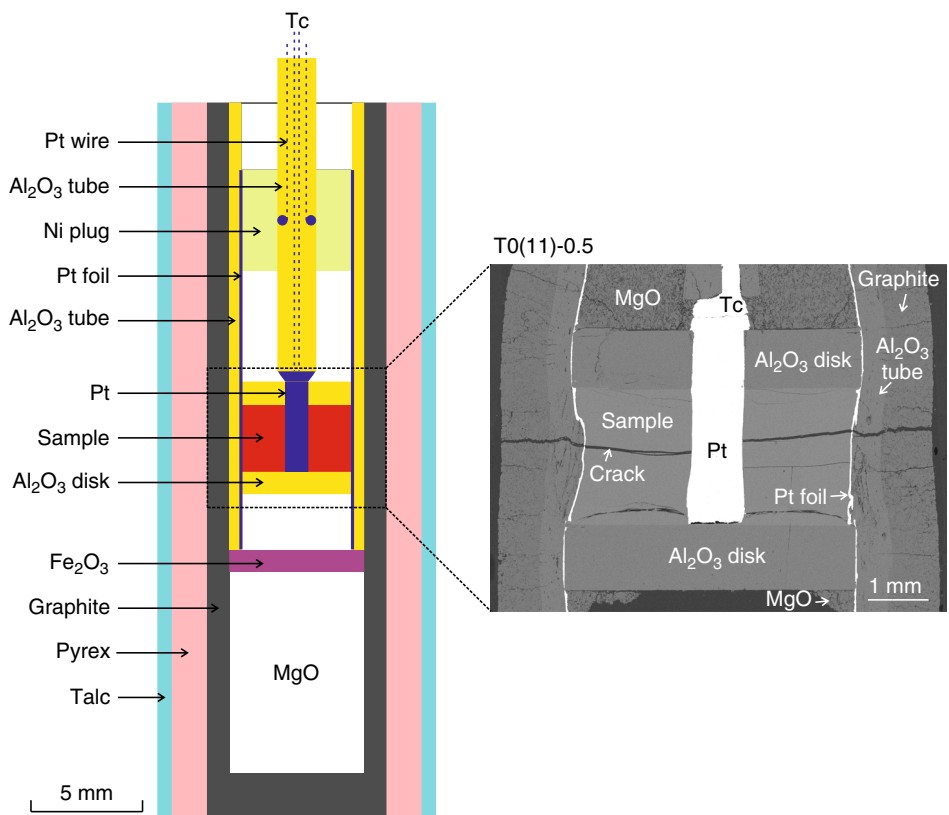

**Fig. 2** Half-inch experimental assembly for piston-cylinder adapted to electrical conductivity measurements. It was modified after Sifré et al.[43] and Laumonier et al.[42]. The SEM image shows the inner structure of the assembly after the experiment on the sample T0(11)-0.5

meteoritic waters, the alternative explanation is that upper crustal fluids are not brines but result from crystallization of granitic magma at depth; this matches the geochemical features[47] of the south Tibetan springs. Partial melting is therefore an alternative explanation to aqueous fluids.

Our laboratory results presented above show that melts can be orders of magnitude more conductive than solid crustal rocks. This implies that the electrical conductivity within the crust correlates with melt fraction. In addition, we see that melt conductivity increases with temperature and water contents and decreases with increasing pressure, i.e., depth, and a given conductivity value can thus be explained by a large number of combinations of temperatures, melt contents, and water contents at a given depth. These parameters are, however, not independent, since petrological relationships tell us that increasing temperature and/or water content causes an increasing degree of melting. This reduces the number of solutions, but the problem remains far from having a unique solution. As shown in Fig. 4, a large range of temperature (Fig. 4a, b), bulk water content (Fig. 4b, c), and melt fraction (Fig. 4c) can yield the conductivity anomalies referred as P1-P2-P3. In order to reduce the parameter space, we propose below a new methodology.

**Laboratory to crustal-scale conductivities**. In previous sections, we have measured the conductivity of rock samples in the laboratory as a function of melt fraction, water content, temperature, and pressure. To apply these results to the interpretation of geophysical models, an additional step is needed, i.e., we must apply now pressure–temperature conditions to predict the expected conductivity. This will allow us to compare our laboratory measurements with conductivity models derived from MT field data. The $P$–$T$ domain of application and the comparison with geophysics-derived models is explained in the next

paragraph. Our analysis, however, makes a number of assumptions that should be clearly stated as follows:

First of all, the conversion of $P$–$T$ path into a melting model requires the use of a thermodynamic modeling using Gibbs energy minimization algorithm (Perple_X[48]). Converting the melting model into electrical conductivity requires the use of Eq. (1), which was calibrated with our experimental data. In addition, a mixing law is required to compute the conductivity of composite materials such as partially molten rocks and this is explained below in the last point.

Second, laboratory experiments use a frequency range of $10^6$–1 Hz, which is higher than those used in MT surveys (100–0.001 Hz). The first difference is not that serious since the impedance spectroscopy uses a variable frequency that allows us to determine the direct current (DC) conductivity of the melt in the laboratory, which is the parameter obtained from MT modeling.

Third, laboratory studies use mm-size samples as opposed to the kilometre-plus scale of in situ rock units or of the MT anomalies discussed here. This difference is a challenge in all applications of laboratory-scale studies to the interpretation of field data. The DC conductivity, however, does not depend on the size of the object so that jumps from the laboratory to crustal scale can be made, which is not the case for other laboratory-geophysical conversions such as ultrasonic-to-seismic wave velocities. What really matters is the size of the unit of rock having a homogeneous distribution of melts and whether this matches the resolution of MT surveys. It, however, remains true that we do not really know how heterogeneous the crust is below Tibet, what is the geometry and density of the conduits containing melts. Broadly speaking, outcrops of migmatite rocks, being often considered as field evidence for the past presence of melt[5,22,24], record the multiple superimpositions of successive

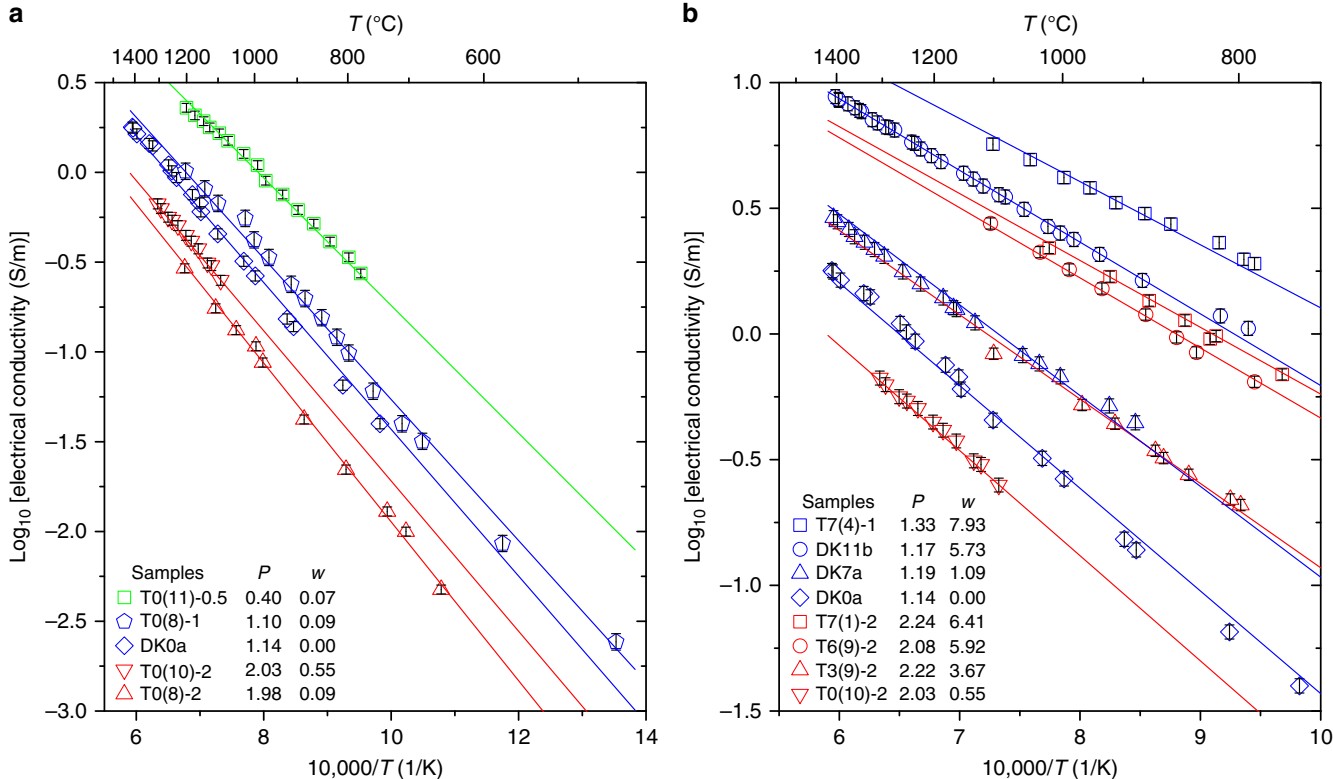

**Fig. 3** Electrical conductivity measurements on dry and hydrous melts plotted in Arrhenius diagrams. The straight lines indicate the fits of the $P$–$T$–$H_2O$ model (Eqs. (1)–(4)). $P$ is the pressure (GPa), $w$ is the water concentration (wt%). **a** Experimental results for nominally dry samples at different pressures. **b** Experimental results for different water concentrations at ~1 GPa or ~2 GPa

melting/tectonic events making tentative the impartial definition of the level of homogeneity and the size of homogenous cells undergoing partial melting at a given time. Such an achievement would require a thorough regional reconstruction involving an exhaustive number of age determinations of rocks and magmatic events that may be impossible to realize. All in all, we assume in the following that, over distances of 500 m, temperatures are broadly homogeneous and this corresponds to the resolution of MT survey at crustal depth[34]. If temperature is homogeneous, assuming that melt remains in the source, the melt fraction and melt distribution from a homogenous lithology must also be homogenous. Heterogeneities at smaller scale may well exist, but our methodology would average the resulting electrical signal.

Fourth, it is also important to understand the uncertainties associated with conductivity models derived from MT data. Converting MT data into a resistivity model is a non-linear process and this means that uncertainties in the measured data cannot easily be converted into uncertainties in the model. For example, in interpreting the high conductivity features in the models the following points should be understood. The MT inversion seeks a model that fits the MT data and is as spatially smooth as possible. MT can define the conductance of a conductive layer, meaning that models with thinner, more conductive layers will also fit the data. Thus generally the conductivity values derived for a conductive layer can be considered the lower limit. Higher values are possible and can only be limited by the laboratory experiments. Others have tried to assign uncertainties to the conductivity of certain model features, but this is very much model specific. Assuming that the model conductivities can be found in the range 30% of the central value is probably realistic.

Finally, the calculation of bulk conductivity also requires that we know the distribution of partial melt with the rock. A variety

of equations have been proposed to model the conductivity of a partially molten rocks and generally produce similar results provided that the melt is interconnected[17,19,43,49]. Melt connectivity in such felsic system is believed to occur >6–7 vol% melt[13], which is below the melt fractions estimated in our calculations. This is due to the topology of the melt stability in this specific system, switching from 0 to ca. 10% of melt with almost no case between 0 and 10% of melt (see Fig. 6 of Hashim et al.[22]). As a consequence, at the onset of melting, the melt fraction rapidly rises to values above melt connectivity threshold[13], which yields high conductivity. The melt conductivity was calculated from Eq. (1) and the rock matrix conductivity was taken from Eq. (3) in Hashim et al.[22] The melt distribution was assumed to be close to the Hashin–Shtrikman upper bound model[49] (Fig.4).

**Electrical conductivity during burial and exhumation of GHS.**
Petrological and thermodynamic constraints allow us to model the conditions encountered during the burial and exhumation of rocks from the GHS by calculating the amount and composition of melt at each point on the $P$–$T$ path. This approach avoids the limitation of previous conductivity measurements made on unequilibrated partial melting experiments[22]. We considered a metapelite with a bulk composition representative of the GHS sequence[2] at the $P$–$T$ conditions of melting[31] in agreement with the G1 $P$–$T$ path (inset diagram in Fig. 5a) predicted by thermomechanical lithospheric models (e.g., Jamieson et al.[18]). The G1 $P$–$T$ path tracks a specific particle through the orogen from 54 Ma to the present[18]; the $P$–$T$ conditions followed by this particle do not necessarily apply to other particles in other places and at other times. As a first-order approach, we assume this G1 $P$–$T$ part, which has been validated by some regional thermobarometric reconstructions[31]. This $P$–$T$ path is used here as a

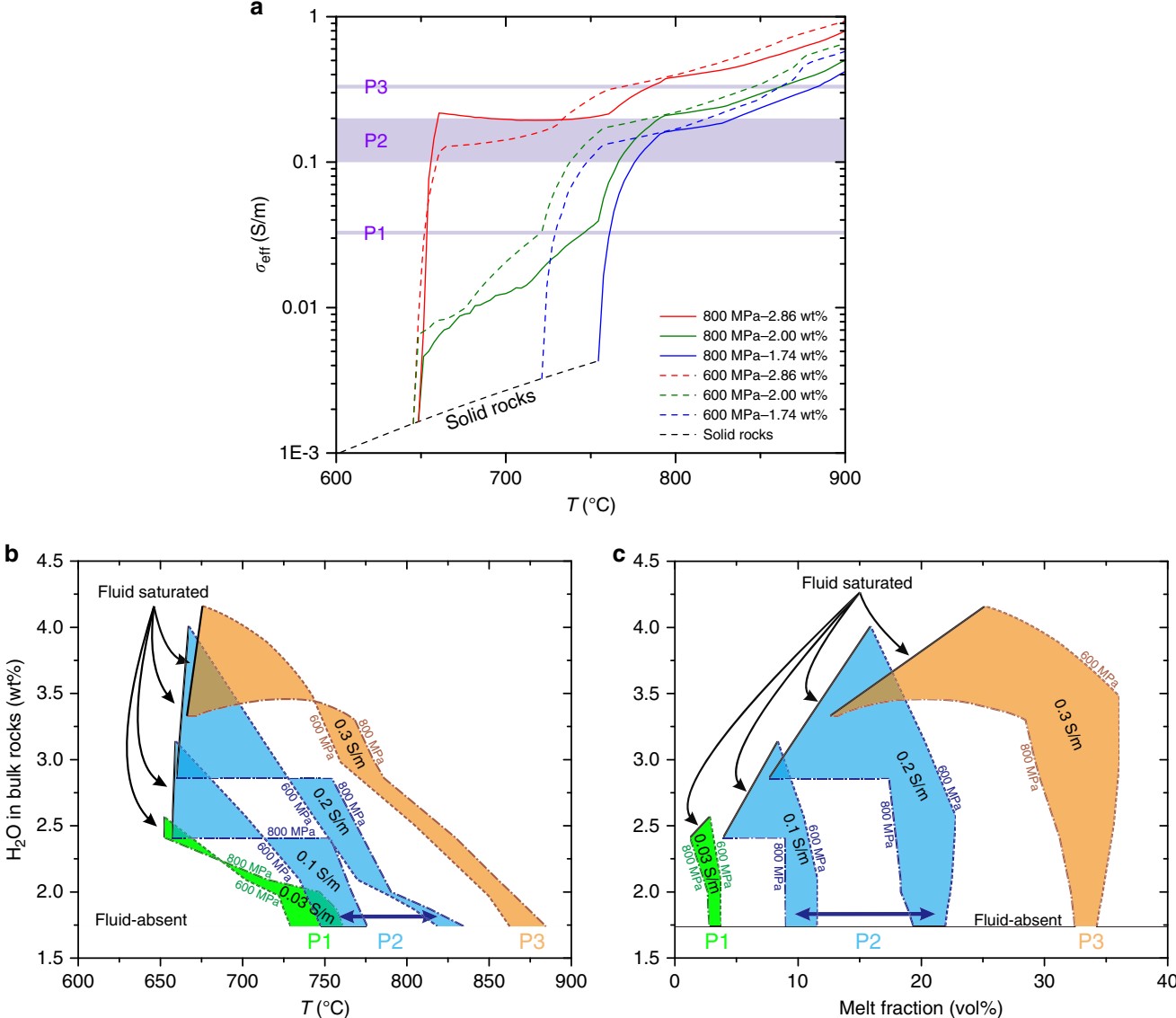

**Fig. 4** Melting, H₂O, temperature, melt fractions, and the multiple solutions explaining the electrical anomalies beneath Tibet. **a** Relationship between $\sigma_{eff}$ and temperature with different H₂O content in the bulk rock at 600 MPa (dashed lines) and 800 MPa (solid lines). The black dashed line refers to the solid rock before melting starts. **b**, **c** Range of temperatures (**b**) or melt fractions (**c**) and H₂O bulk rock contents satisfying the conductivity of 0.03 S/m (P1), 0.1–0.2 S/m (P2), and 0.3 S/m (P3), respectively. The boundaries of fluid-absent and fluid-saturated magmas identified by black straight lines confine the area of possible solutions satisfying the conductivity values previously defined; in between these two boundaries, the regime of fluid-present melting prevails. A single value of conductivity therefore corresponds to a large range of solutions in the $T$-melt content–H₂O content space. All these parameters are, however, related by petrological relationships that can be embodied into a geodynamic scheme such as the G1 $P$–$T$ path (Fig. 5)

reference thermal framework since it has the great advantage of being corroborated by thermobarometric studies on exhumed Miocene rocks.

Two water contents are distinguished: A first case considers 1.7 wt% H₂O in the source (fluid-absent melting) and a second case considers 2.9 wt% H₂O in the source (fluid-present melting). This first case corresponds to the storage capacity of micas at subsolidus conditions. The second case permits a small amount (1.2 wt%) of excess H₂O in addition to the water stored in micas. This does not mean, however, that an aqueous fluid phase remains present during the entire melting course.

As shown in Fig. 5, most of the prograde part of the G1 path occurs at subsolidus conditions (see the lines AB and CD in the inset diagram). The onset of melt stability seems to coincide with the near-isothermal decompression marking the beginning of rock exhumation. Melting stops during cooling and

decompression along moderate d$P$/d$T$ retrograde paths, mainly in the sillimanite field[18,24,31].

Both fluid-present[29,50] and fluid-absent[24,31,51] melting are considered in Table 3 and in Fig. 5. For the case of fluid-absent melting, the first liquid appears at 793 °C–1.31 GPa along the prograde path and the solidus is crossed at 715 °C–0.57 GPa during the retrograde stage (Fig. 5). Fluid-present melting occurs at lower temperatures (<650 °C) reached at 1.22 GPa and produces more melt (15–25%), making the system more conductive, for a given temperature (Fig. 5).

The prediction of Fig. 5 can first be applied to the northwestern Himalaya (P1, P2, and P4) as demonstrated below (see also Table 3). Based on our calculations, we infer that fluid-absent melting along the G1 $P$–$T$ path (red curves in Fig. 5) can explain the electrical conductivity anomalies P1 and P2 very well (see Fig. 6 for a petrological and spatial framework localizing P1–P5

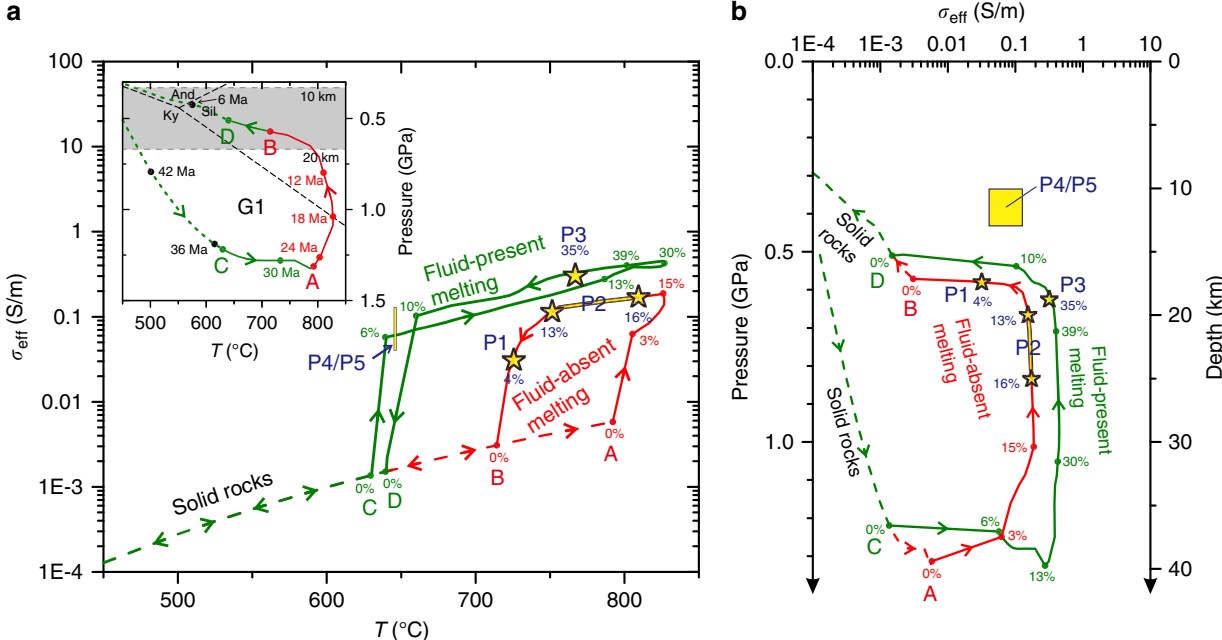

**Fig. 5** The electrical conductivity ($\sigma_{eff}$)–pressure–temperature paths during burial and exhumation of Miocene Himalayan rocks compared with the conductivity of present-day Himalayan crust derived from MT data. **a** $\sigma_{eff}$ along the *P–T* G1 path (inset diagram) as predicted by the "channel-flow model HT1"[18] and metamorphic petrology[31]; the numbers in Ma on G1 path indicate the particle's ages to the present. **b** Electrical conductivity vs. pressure or depth along the G1 path. The red path represents melt in the absence of excess water (i.e., fluid-absent melting, approximately dehydration melting) while the green path illustrates melting in the presence of excess water (i.e., fluid-present melting with excess water content of ca. 1.2%). Dashed curves indicate solid rocks (below solidus), while solid curves (i.e., the lines AB and CD) identify the partially molten domains. Green lines indicate fluid-present melting (AB), while the red lines refers to fluid-absent melting (CD). Arrows represent the direction followed during prograde and retrograde metamorphism along the G1 *P–T* path. The yellow stars and rectangles indicate the conductivity at P1–P3 and P4/P5 (see Fig.1b, c) from the MT studies[17,34], respectively, and their sizes represent 30% uncertainty on the MT results. The percentage values on the curves represent the equilibrium melt fractions along G1, and the blue labels define the melt contents at P1–P3 (see also Table 3). The high conductivity spots (P1 and P2) are consistent with fluid-absent melting while P3 may correspond to fluid-present melting; alternative explanation for P3 are a local melt accumulation process or temperature being 100 °C warmer than those of the G1 *P–T* path (see text)

**Table 3 Depth and magmatic conditions of the P1–P5 anomalies from northwestern Himalaya and southern Tibet**

| Pos | Depth (km) | $\sigma_{eff}$ (S/m) | P (GPa) | T (°C) | Melt content (vol%) | Melt water content (wt%) | Magmatic process |
|---|---|---|---|---|---|---|---|
| *Northwestern Himalaya* | | | | | | | |
| P1 | 15–20 | 0.033 (0.023–0.043) | 0.58 | 726 (723–729) | 4 (3–5) | 7.8 (7.6–7.9) | Fluid-absent melting |
| P2 | 20–25 | 0.11 (0.08–0.14) | 0.60 (0.59–0.63) | 750 (739–773) | 13 (9–16) | 7.1 (6.6–7.4) | Fluid-absent melting |
| | | 0.17 (0.12–0.19) | 0.82 (0.60–1.01) | 812 (754–827) | 16 (15–17) | 6.5 (6.2–7.2) | |
| P4 | 10–13 | 0.05 (0.04–0.07) | 0.40 | 645 (645–646) | 20 (10–30) | 8.3 | Crystallization |
| | | 0.10 (0.07–0.13) | 0.40 | 647 (646–647) | 20 (10–30) | 8.3 | |
| *Southern Tibet* | | | | | | | |
| P3 | 18–20 | 0.33 (0.23–0.43) | 0.62 (0.59–0.88) | 767 (737–818) | 35 (27–39) | 6.8 (6.2–7.5) | Fluid-present melting (+1% $H_2O$) |
| | | 0.33 (0.23–0.43) | 0.62 | 864 (839–885) | 34 (24–42) | 3.9 (3.5–4.6) | Fluid-absent melting (out of G1) |
| P5 | 10–13 | 0.05 (0.04–0.07) | 0.40 | 645 (645–646) | 20 (10–30) | 8.3 | Crystallization |
| | | 0.10 (0.07–0.13) | 0.40 | 647 (646–647) | 20 (10–30) | 8.3 | |

The 30% uncertainty of electrical conductivity anomalies ($\sigma_{eff}$) was considered and propagated in terms of temperatures, melt fraction, and melt water content (marked in parentheses next to each value) maintaining the G1 *P–T* path. Note that two values are considered for the P2 anomaly (following ref. [34]) and two solutions are proposed for P3 (fluid-present melting along G1 vs. fluid-absent melting under warmer conditions)

anomalies). P2 anomaly contains 13–16 vol% of melt at a temperature of 750–812 °C and a depth of 20–25 km (i.e., retrograde *P–T* path of G1). These melts contain 6–7 wt% $H_2O$. The P2 anomaly likely reveals the uppermost point of the isothermal exhumation of the GHS, where the system actually starts to cool down more rapidly as it rises up. The P1 anomaly is well explained by the nearly isobaric cooling stage of G1,

involving the crystallization of a large part of the melt that has been produced deeper, i.e., at the P2 stage; at the depth of P1 or deeper, the melting model for path G1 predicts 4 vol% remaining melt containing ~8 wt% $H_2O$ at ~726 °C.

A similar analysis can be made for southern Tibet (i.e., points P3 and P5, see Fig. 6). The P3 anomaly (~0.3 S/m) is slightly but significantly more conductive than the $\sigma_{eff}$-depth path

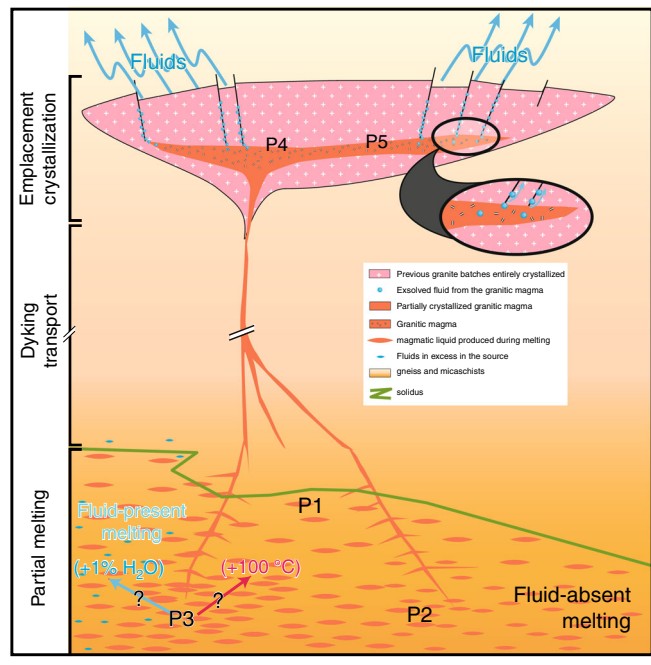

**Fig. 6** Illustration linking P1–P5 MT anomalies to magmatic processes and crustal-scale transfer of water. The magmatic conditions at the P1–P5 electrical anomalies are given in Table 3. By these magmatic processes, water is transferred from the lower crust (depth >20 km), where partial melting occurs, to the emplacement/crystallization zone of granites (depth = ca. 10–12 km). Aqueous fluids released from solidifying granites may well significantly contribute to the numerous hot springs suggested from their geochemical features[47]. Fluid-absent melting under conditions similar to the G1 *P–T* path can explain the Northwestern Tibet, while beneath the Southern Tibet, either fluid-present melting or a warmer crust is required

corresponding to partial melting along the G1 *P–T* path, even considering the 30% uncertainites (0.23–0.43 S/m) in the MT data. P3 can, however, be explained by ~35 vol% of interconnected melt containing ~7 wt% $H_2O$ at ~767 °C (green curves in Fig. 5), which would be produced for fluid-present melting along the G1 *P–T* path. Here we considered 2.9 wt% bulk water content in the source, i.e., 1.2 wt% of excess $H_2O$ (i.e., "free water") in addition to 1.7 wt% $H_2O$ stored in hydrous micas (see Hashim et al.[22]). If the system contained more water, the electrical conductivity of P3 could be explained with a lower melt fraction at a lower temperature. For the sake of simplicity in our discussion and to remain consistent with the range of temperatures and water contents determined by petrological constraints[2,41], we assumed a *P–T* path similar to G1 underneath southern Tibet and north-western Himalaya. The P2 anomaly then must then reflect fluid-absent melting beneath the north-western Himalaya while the P3 anomaly would reflect fluid-present melting beneath southern Tibet, with about 1% excess aqueous fluid at the onset of melting (Fig. 6). We must admit that, though this is a matter of debate[29], fluid-present melting is generally not considered in the Himalaya[2,24]. Here we suggest that fluid-present melting is required to explain P3 because we assume the G1 *P–T* path, but alternatives can be considered: A warmer geotherm, with temperature of about 864 °C (Table 3, Fig. 4) at ca. 20 km depth would produce the appropriate conditions yielding a conductivity similar to that at P3 in the fluid-absent regime. This is ~100 °C warmer than the G1 and warmer than any *P–T* estimate in the Himalaya, but it is not strictly inconsistent with some results of thermal modeling

obtained in Jamieson and Beaumont[52]. Interestingly, such a configuration would produce melts being relatively poor in water (<4 wt%) with a reduced buoyancy. At this stage, we can therefore propose that the P3 domain is compatible with the *P–T* conditions of the G1 path with the local presence of small quantities of hydrous fluid at the onset of melting (1%) or may be produced by a locally warmer geotherm (i.e., by ~100 °C). So far, we consider that no sound constraints could exclude the local occurrence of fluid-present melting with 1% of excess fluid at the onset of melting while a warmer crust in the southern Tibet can neither be excluded. Both scenarios could fit the geodynamic background given that Jamieson and Beaumont's[52] thermal models allow present-day conditions being warmer than the G1 at ca. 20 km and that the Indian basement that is buried beneath the Tibet could provide the small quantity of fluids being required to locally produce fluid-present melting conditions (Fig. 7). At this stage, only the construction of a new generation of thermomechanical models considering thermal, petrological, and geophysical inputs could help discussing further these cases.

**Melt migration, dykes and plutons**. Figure 7 suggests a continuous evolution from the Miocene to the Present with the Miocene migmatites and plutons being solidified fossils of the present-day partial melting zones sampled by MT soundings. This may seem an oversimplification as the STD and the MCT had ceased to move by ~11 Ma. However, at the depth of partial melting, the MCT and the active MBT and MFT merge into a single larger-scale structure, the Main Himalayan Thrust (Fig. 7),

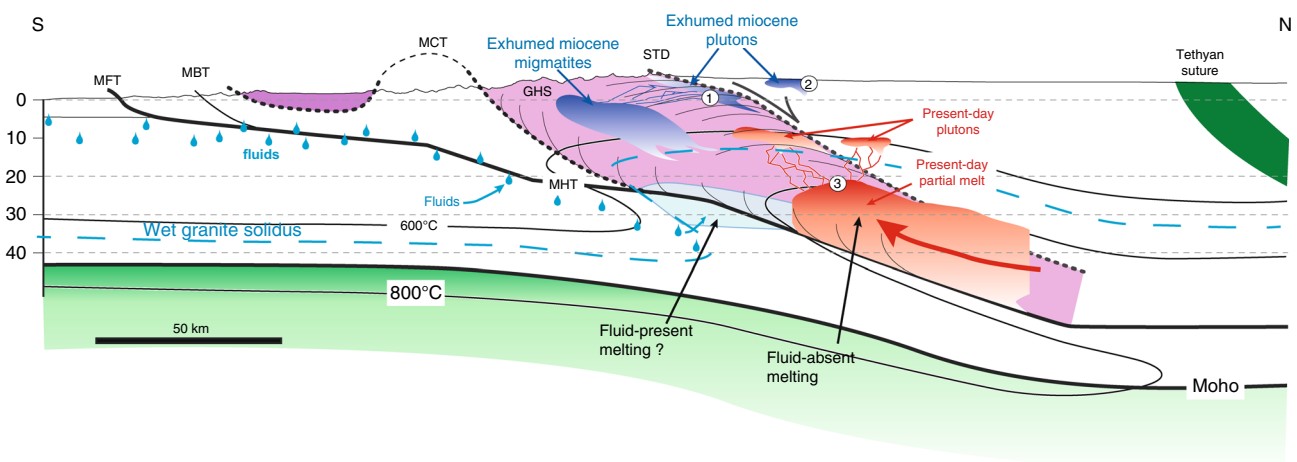

**Fig. 7** Schematic diagram of granitic magma production, migration, and emplacement in a schematic cross-section of the Himalaya. The inactive shear zones (i.e., STD and MCT) are shown with bold dotted lines, while the active shear zones (i.e., MBT, MFT, and MHT) are shown as bold straight lines. The red regions identify the present-day location of partial melting plus melt ponding (present-day leucogranite plutons) while the blue areas are exhumed and solidified granitic rocks (migmatites and plutons). The numbers illustrate the emplacement of granitic plutons (1) below the STD or (2) above the STD while (3) refers to the top of the partial melting region from where the present-day dykes feeding the growing plutons are nucleating. The fluid identified by Rawat et al.[40] is delineated by the blue droplets, south of the Main Frontal Thrust, and located within the Indian basement. Melting is obviously not occurring in the cold regions of the Indian basement, but this must represent a fluid-enriched zone that is buried beneath the chain. What happens to these fluids is not the kernel of this study but they may well contribute to the process of melting beneath Tibet

and the basal conditions have thus not changed significantly. The cessation of motion along the STD has since changed the mechanism of overburden removal, but some less localized active mechanism involving erosion is at work anyway, making our scenario possible.

By interpreting the conductivity anomalies in terms of melt fractions predicted from the model $P$–$T$ path during retrograde metamorphism, we assume that the decompression and cooling associated with exhumation involve melting and crystallization in a closed system, respectively, i.e., the melt produced does not migrate away from its source and/or does not locally accumulate. We calculated melt flow velocities for melt contents ranging from 5 to 35 vol% by Darcy-type flow (percolation) to be << 1 cm/year (see Methods for more details), which is the denudation rate of the GHS assumed by Beaumont et al.[12]. Pervasive melt migration along grain boundaries therefore appears to be a relatively inefficient mechanism to effectively move and accumulate >30% melt at a scale visible by MT. In contrast, the excess pressure (i.e., the melt pressure minus the solid pressure[53]) reaches 17 MPa for the case of P3 (35 vol% melt) and 9 MPa for the case P2 (16 vol% melt) (see Methods for details). Such values of melt overpressure are in the expected range (1–40 MPa)[53] to break the surrounding crust and allow dykes to nucleate. These dykes most likely feed leucogranite plutons located at higher structural levels[30] and therefore contribute to the removal of melt from the upwelling of partially molten regions (Figs. 6 and 7). We must specify here that this threshold overpressure is poorly known and should change with temperature, stress level/deformation, and pre-existing mechanical weaknesses[53,54] but we suggest that the inferred melt contents (e.g., 16% and 35%) represent the maximum melt content that the local system can sustain before melt removal by diking.

Leucogranite plutons sometimes reach a size of tens of kilometres and they are believed to be fed by such dykes[30]. According to our experimental data, the occurrence of sizeable, largely molten plutons would produce very high conductivity (>1 S/m) at a depth of ca. 10–12 km, which is the depth at which these plutons grew during the Miocene. However, such shallow regional electrical anomalies have not been observed in MT

campaigns since the P4 and P5 anomalies (at depth of 10–12 km) are 10–50 times less conductive than what should be expected from a fully molten granite containing 6–8 wt% $H_2O$. The absence of geophysical signals revealing largely molten plutonic bodies beneath Tibet corroborates thermal modeling coupled to geochronological constraints[55] predicting that these plutonic bodies cannot be fully molten at any given time but instead are the result of the slow amalgamation of small melt batches over several My. Combining the $T$–$X_m$–$H_2O$ of crystallization of leucogranites from Scaillet et al.[41] with the effect of crystals on the electrical conductivity from Gaillard and Marziano[49], the electrical conductivity of cooling and crystallizing leucogranitic magmas was calculated. This allowed us to suggest that the magnetotelluric signals at a depth of ca. 10–13 km[17,34] indicate plutons growing beneath the Tibetan plateau. Those plutons are mostly solid and contain a maximum of $20 \pm 10$ vol% of leucogranitic melts (containing ~8 wt% $H_2O$) at $645 \pm 5$ °C. This corresponds to the conductivity of P4 in northwestern Himalaya and P5 in southern Tibet (Fig. 1b, c, Table 3, Fig. 6). The position of these hypothetical actively growing plutons is north of the two leucogranite belts that outcrop in the GHS and the THS (Fig. 1a).

**Weakening and water transfer through the crust.** The methodology we develop here allows us to link the geophysical MT data to the petrology of crustal melting, being ruled by the present-day thermal state of the Himalayan crust. This allows us to discriminate regions of melt residence such as migmatites and regions of pluton emplacements (Figs. 6 and 7). We can also infer the melt fractions and melt water contents and relate this to the thermal state prevailing in the Tibetan crust some Ma. Furthermore, this approach also allows us to suggest either melting enhanced by a small amount of fluids or warmer thermal regime in the southern Tibet linking geophysical with either geochemical or thermal anomalies. In most cases, the melt content of the rocks constituting these large geophysical anomalies (15–35%) significantly exceeds melt connectivity threshold and weakens crustal rocks[13]. This conclusion is at odds with the recent assessment for the Qiangtang terrane, suggesting that only ca. 2%

melt is needed to account for the crustal seismic properties[56]. Though the seismic and electrical anomalies are spatially coincident (Supplementary Table 1) and occur in the depth range at which partial melting is expected to occur[24], this discrepancy in the assessed melt fraction is unexplained. Establishing quantitative relationships between melt fractions and seismic velocities remain a difficult task (ref. [57]) and it is clear that 2% of melt would not produce the conductivity ranges that are deduced from the MT surveys on the Tibetan crust; about 10 times more melt is needed. The viscosity corresponding to such regions is defined by laboratory studies to be of the order of $10^{11}$ Pa·s[13,22]. These regions of high conductivity must then represent long-lasting weak zones that play a key role in the initial development and longevity of orogens (see Fig. 7). The link between this region of crustal melting and the northward broad zone of mantle-derived magmatism[15] remains tentative and beyond the purpose of our study, though there may well be a geophysical continuity[17].

All in all, whether fluid-absent or fluid-present melting is involved, the magmatic processes triggering such electrical anomalies indicate massive amounts of water involved in the partial melting process, transferred by the melts as dykes and released by solidifying granitic bodies forming the plutonic rocks (Fig. 6). This must provide fluids at the base of the upper crust (liberated from solidifying granites at ca. 12 km, 600 °C). These fluids must contribute to the numerous hot springs that feature the South Tibet as suggested by their helium, carbon, and nitrogen isotope compositions[47] being consistent with the metapelitic source of granitic rocks.

## Methods

**Sample syntheses**. Samples were first crushed into a fine-grained powder and fused twice at 1600 °C and 1 atm to produce a homogeneous, volatile-, and bubble-free glass. This glass was crushed again and directly used for anhydrous experiments. Hydrated glasses were prepared by equilibrating the dry glass powders with the desired amount of water in a welded-shut gold capsule at 400 MPa and 1000 °C in an internally heated pressure vessel for 3 days. These hydrous syntheses were then crushed into fine-grained powders to provide the starting material for conductivity measurements on the hydrous samples.

**Experimental processes**. The experiments began with systematic cold pressing of the glass powder to ~3 GPa in piston cylinder for 10 h. The temperature was quickly increased to ~700 °C in <2 min, followed by minor adjustments of the target pressure. Next, one heating–cooling cycle was applied to the sample and run durations were kept <90 min to minimize crystallization and water loss. A few samples significantly crystallized (not shown here) during the experiments (mostly dry samples exposed to high temperature and pressure) and were discarded. The sample resistance was measured by alternating current impedance spectroscopy during the heating and cooling cycle (Supplementary Fig. 3). The calculation of the conductivity measurements follows refs. [42,43].

**Measurement of the water concentration**. The water concentration of glassy samples was determined by FTIR at ISTO before and after each experimental run (Supplementary Fig. 2). Double polished chips of glasses with a thickness of 200–300 μm were analyzed. The Beer–Lambert law was used to calculated the water concentration using the extinction coefficient of Ihinger et al.[58]. In general, the fundamental $H_2O$-stretching vibration (3530/cm) was used to determine the water concentration of anhydrous samples (Supplementary Fig. 2b) while the molecular water (5200/cm) and $OH^-$ (4500/cm) stretching vibrations were used to determine the concentration of hydrated samples (Supplementary Fig. 2a). FTIR analyses usually give a precision of about 10–15% on the water concentration. Additional details on the production of hydrated syntheses and their FTIR analyses can be found in Laumonier et al.[42].

**Chemical composition and water concentration**. The chemical composition of quenched glasses after experiments is identical to the starting material (Supplementary Table 2). For the water concentrations of experimental samples, the dry samples [T0(11)-0.5 and T0(10)-2] remains "nominally dry" after experiments (up to 0.06 and 0.55 wt% $H_2O$, the water coming likely from the talc used in the assemblies), and the water content of the glass after the conductivity experiments using the T12-22 starting composition (experiments named T0, T3, T6 and T7) is similar to that before the conductivity measurements. In contrast, long duration experiments on the starting material DK89 suffered from a substantial loss of water (Table 2).

**Modeling the electrical conductivity of melts**. We fitted the entire set of conductivity data using Eqs. (1)–(4) in order to adjust the $P$–$T$–$H_2O$ dependencies of the electrical conductivity. During this inversion, we allowed the experimental pressures and water concentrations to vary within an optimization range; this range includes the uncertainties and the variabilities, i.e., we applied a ±10% interval to an associated nominal pressure (i.e., uncertainty plus variability during a $T$ cycle) and the water concentration was considered to be in the interval of water concentrations measured before and after the conductivity experiments, these end members being characterized by an uncertainty of ±15%. The resulting $P$ and $H_2O$ contents are labeled "adjusted $P$" and "adjusted $H_2O$" in Supplementary Table 3. The correlation coefficient associated with the model of electrical conductivity of leucogranites is 0.98. The error bars of each measurement placed on Fig. 3 result from the propagation of uncertainties in sample geometry (including the thermal expansion during melting) and ~5% precision of the resistance measurements. The average error value of conductivity is 0.03 log units and it compares well with the model that we proposed (see Eqs. (1)–(4) in the main text) having an uncertainty of 0.04 log unit in conductivity. After calculation from the Nernst–Einstein equation, sodium can be regarded as the dominant charge carrier in the leucogranitic melt (See Supplementary Note 1 and Supplementary Fig. 5). The resulting model is compared with the other studies including the recent work on peralkaline silicic melts[59] in Supplementary Fig. 6 implying that our $P$–$T$–$H_2O$ model (Eqs. (1)–(4)) can be used for a broad range of melt compositions formed by partial melting of the crust and in various settings[60] (Supplementary Note 2).

**Equilibrium fractions and water contents of melt**. Using the collection of thermodynamic modeling program Perple_X[48], equilibrium melt fraction and melt water content were calculated as a function of $P$ and $T$ by creating a $P$–$T$ and $T$–$X$ ($H_2O$) pseudosection following Hashim et al.[22]. Calculations were performed using a biotite-muscovite metapelite from the Sioule valley (France) (see Hashim et al.[22], Supplementary Table 4). in a NCKFMASHT system, using the thermodynamic database of Holland and Powell[61] and solid solutions from Coggon and Holland[62], Newton and Haselton[63], and White et al.[64,65] $P$–$T$ pseudosections were produced between 0.2 and 1.2 GPa from 600 to 900 °C with a bulk water of 1.7 wt.% $H_2O$ (fluid-absent melting) and 2.9 wt% $H_2O$ (fluid-present melting). $T$–$X$($H_2O$) were calculated at 0.6 and 0.8 GPa from 600 to 900 °C and $H_2O$ contents from 0.1 to 5 wt% (Fig. 4).

**Electrical conductivity during partial melting**. The electrical conductivity ($\sigma_{eff}$) of metapelites undergoing partial melting along isobaric (600 and 800 MPa) heating path was calculated (Fig. 4a). The results suggest that decreasing pressure and the presence of minute amount of aqueous fluid (~0.3 wt%) can decrease the melting temperature down to ~ 650 °C by promoting melting of plagioclase and quartz[2]. At the onset of melting, $\sigma_{eff}$ increased quickly due to high $H_2O$ content in the melt and due to the fact that such a system shows an abrupt slope in the melt fraction vs. temperature ($T$) relationship; for higher degree of melting, $\sigma_{eff}$ increases relatively gently as melt fraction increases and $H_2O$ gets diluted in the melt.

In Fig. 4b, c, we represented the range of $T$–melt fraction–bulk $H_2O$ content in the pressure range 600 to 800 MPa satisfying the conductive spots of P1–P3 that are located on Fig. 1b, c and whose conductivity values are given in Fig. 4a. Each colored area represents the $T$–melt fraction and $H_2O$–melt fraction domains matching P1–P3 (from the less to the most conductive spots). Each area is lower bounded by the onset of melting and upper bounded by the saturation of the melt in $H_2O$ (i.e., maximum water content in the melt returned by Perple_X). We see that the most conductive anomaly, P3, can be explained by fluid-absent melting producing >30% melt at $T$ ~ 875 °C and by fluid-present melting at $T$ < 700 °C and melt fraction <20% if we consider a bulk $H_2O$ content of ca. 3.5 wt%. The melt produced under such conditions would, however, contain a lot of water (>10 wt%) therefore forming at lower temperatures than the G1 $P$–$T$ path requires, which is not fully consistent with available petrological constraints on the composition and $P$–$T$–$H_2O$ conditions of crustal melts in the Himalaya[2,41]. These petrological constraints indicate melt water contents in the range 5–7 wt% and temperatures of 750–800 °C at the locus of melt production. We prefer here considering a unique $P$–$T$ path (Fig. 5) that satisfies (1) the $P$–$T$ path retrieved from ancient exhumed rocks from the Himalaya[31], (2) thermomechanical models of the extrusion of the GHS, i.e., the G1 $P$–$T$ path after Jamieson et al.[18], and (3) the $P$–$T$ conditions of crustal melting and leucogranite production deduced by petrologic constraints[2].

**Calculation of melt velocities and overpressure**. In the absence of compaction, the Darcy velocity of the melt can be derived from the expression of force balance in a two phase mixture[66] and it reads

$$\Phi\left(v_f - v_s\right) = \frac{k(\Phi)}{\eta_f}\delta\rho g \tag{5}$$

where $\Phi$ is the melt fraction, $\eta_f$ is the melt viscosity (Pa·s), $k(\Phi) = \frac{d^2}{C}\Phi^n$ is the crust permeability ($m^2$) with the parameter $C$ and $n$ are bracketed by experimental values[67], $d$ is the grain size (see below), $\delta\rho = \rho_s - \rho_f$ is the density gradient between rock and melt, and $g$ is the acceleration constant ($m/s^2$). Viscosity and density of leucogranite melts were taken from Scaillet et al.[68].

When compaction is taken into account, compaction waves develop and form melt-enriched zones with a size similar to the compaction length[69]:

$$\lambda = \sqrt{\frac{\eta_s k(\Phi_0)}{\eta_f \Phi_0}} \qquad (6)$$

where $\Phi_0$ is the background or melt-free porosity and $\eta_s$ is the solid crust viscosity (Pa·s). A first-order approximation of the overpressure within these melt-enriched zone is given by $\delta P = P_f - P_s = \lambda g \delta \rho$[70].

**Estimation of grain sizes of the GHS metapelite**. In addition to the estimation of the grain sizes given in Groppo et al.[31], two gneisses collected in the GHS (Purang and Nyalam county, see Fig. 1a) were analyzed by microphotographs. The grain sizes where partial melting occurred range from 0.3 to 0.7 mm.

## Data availability
The authors declare that the data supporting the findings of this study are available within the paper and its supplementary information files.

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

## Acknowledgements

Part of this study was financially supported by the Special Fund for Earthquake-Scientific Research in the Public Interest (Grant no. 20150818), the National Natural Science Foundation of China (Grant nos. 41372202 and 41672197), while the experimental work was supported by the European Research Council (Grant no. 279790). J.C. acknowledges the China Scholarship Council (CSC) for a visiting scholarship to Université d'Orléans, Institut des Sciences de la Terre d'Orléans (ISTO). A.V. acknowledges funding from Labex VOLTAIRE (ANR-10-LABX-100-01) and the ANR program VARPEG (ANR-15-CE01-0001). We thank Qingbao Duan and Lining Cheng for providing us samples of leucogranitic bulk rocks and the thin sections of gneisses from Purang county and Nyalam county in Tibet of China P.R., David Sifré for helping us with the experiments of electrical conductivity measurements, and Giada Iacono Marziano and Ida Di Carlo for analytical assistance.

## Author contributions

F.G. led the project. X.Y. supported the field work in Tibet of China P.R. J.C., F.G., and M.L. developed the experimental set-up. J.C. and M.L. performed the conductivity measurements. J.C. conducted sample syntheses, textural, and chemical observations. A.V. calculated equilibrium melt fractions and melt water contents. J.C. and F.G. wrote the first draft and all authors, A.V., X.Y., M.U., L.J., L.H., B.S., and G.R. contributed equally to the writing of subsequent drafts. J.C. produced Figs. 1–5, A.V. produced Fig. 6, and L.J. produced Fig. 7.

## Additional information

**Competing interests:** The authors declare no competing interests.

