## [Peer Review File · Nature Communications]

Reviewers' comments:

Reviewer #1 (Remarks to the Author):

Review of "Long-lived weak regions within the Himalayan crust due to water-rich granitic melts"

Reviewed by Susan Ellis, September 18th 2017

Overall impression: In general I liked this paper and thought it was a useful contribution to the debate about melt vs. fluids in Himalayan tectonics. My main thought is that the paper did not "fess up" quite enough about a primary assumption that it makes- and which is critical to support the paper's conclusions. This is the assumption of a kind of "quasi steady-state" (mentioned right at the end of the paper on line 316). That is- that the present-day geophysical anomalies- particularly resistivity- can be related to the rock record from exhumed leucogranites. This assumption is made despite the fact that the main faults bounding the exhumed section (the MCT and STD) are currently inactive. I think the authors need to be a lot clearer about this assumption and to justify it more. They also need to spend more time in the introduction detailing the bounding parameters for activity on the thrust faults (timing vs. tectonics; role in exhuming the partial melt; when did activity cease on these faults?). Provided this is clarified I think the paper will make a nice addition to the literature.

Some minor comments:

1. fluid-absent vs. fluid-enabled terminology and explanations: I got rather confused by the loose way these terms were bandied around in the paper. I think the authors need to clearly define, near the beginning, what these terms mean- for resistivity vs. melting. For example, does fluid-enabled include cases where hydrous minerals are present- but no free water in pore spaces? How do these terms relate to the term w (wt%), and the term "water concentration (wt%" used in Figure S6)? And is w the same as label "H₂O in rocks" on figure S7? What particularly confused me was the sentence on line 213, which claimed that "these melts" (referring to the P2 anomaly, which is claimed to have "fluid-absent melting along the G1 P-T path (line 209)) contain 6.5-7.1% wt% H₂O. Is this a typo? Surely these are then not fluid-absent? I was also confused by lines 222-223 where the % is separated into hydrous mineral water and additional fluid. And how does this relate to the "Adjusted H₂O" also mentioned in the manuscript. In figure caption 3, it says "(i.e. excess water content of ca. 1.2%)"- where did this come from? Is this free water?
2. Figure 1 labels "present day"- not sure what this means as it is not explained in the figure caption
3. Figure 3 does not plot P4 and P5 on the P-T paths
4. Line 287. Please tell us what the extrusion rates are for comparison
5. line 299. "Leucogranite plutons...for such dykes". Are the two parts of the sentence referring to the same thing- are they plutons or dykes?
6. line 315-316. This sentence seems self-contradictory- see main comment above
7. English grammar- in some places it needs some work eg "bear witness of" line 93- to rather than of?
line 172 "very well" sounds clumsy and not needed
lines 208-235. I found this whole section very confusing and at the end of it I was even more confused which anomalies are supposed to be fluid-enabled vs. dry. I suggest a careful rewrite including wt% thresholds used to define the terms and mineral bound vs. pore fluid mentioned.
lines 258-259- meaning not clear, rewrite
lines 266-267- ditto
line 269 "that fluids" perhaps say "that significant fluids"

Methods section

lines 570, 572 "On one side...on the other side" correct the grammar

line 574: "On the contrary"-> "In contrast"

line 597: "were realized" -> "were performed"

line 602: here is yet another definition of fluid-absent vs. fluid-present. Perhaps the confusion arises because the effect of fluids on melt vs. resistivity involved different thresholds? If so clarify this somewhere.

line 632: "with the parameter C and n"-> "Where parameters C and n"

Reviewer #2 (Remarks to the Author):

Overall, I like the idea that the authors are advancing. Linking ancient rock record to modern geophysical data through experimental constraints. This has great potential to allow us to link melt process to product in a more quantitative manner.

However, in my view, the paper falls well short of the impact required for Nature. There are numerous issues with the paper that prevent me giving it my support. Below I briefly summarize my concerns.

Figure 4, in my mind, encapsulates the author's fundamental misunderstanding about the petrology and geochemistry of the granites exposed in the Himalaya, and how, if at all, they might relate to modern day geophysical data. The authors imply, based on the way that draw their figure, that there should be two types of granites exposed in the GHS (fluid-present and fluid-absent). The geochemistry of these rocks is distinct, and is easy to separate out. However, the fundamental problem is that inside the main exposure of the GHC, there are no rocks that were formed by fluid-present melting reactions. No single paper has reported such rocks. Sure, such rocks exist to the north of the main GHC, in domes, and further north (and some way to the south), but none are inside the GHC, as drawn in the cross-section. I raise this point, because it is fundamental to the authors assertions about how melts form and how the Himalaya actually work.

In addition, Figure 4 shows a series of fluids entering the GHC from the subducting Indian plate. This may be feasible (setting aside the likely impermeability of the major bounding fault, HMT), but then, wouldn't fluid-present melts dominate the GHC? Why do such melts not dominate the geologic record inside the GHC, as exposed today?

Finally, the authors assert that melts are able to cross the STDS. Time and again, such observations have either proven to be inaccurate (Hodges in Everest), or that they are in fact beneath the main STDS (Carosi in W.Nepal). The timing of movement, now very well constrained, does not permit melts to cross the STDS.

There is some major confusion with terminology and existing literature that results in the meaning of the ms. being lost in several key places.

Line 61: "granite-related migmatites and granulites"

The authors are confused. A granulite does not by its definition necessarily involve partial melting, or extraction of melt from a rock.

62: "but it is always difficult to distinguish the timing of individual events"

This is simply untrue. The authors ignore many ten's of papers that report high precision, high resolution ages, and for granitic systems, particularly in the Himalaya. There is a vast body of knowledge of this topic that is entirely ignored here, and elsewhere in the literature. As but one example, Lederer et al., 2013, CMP, report numerous high resolution isotope geochronology ages of granites from NW India. These authors were able to construct a model, that is very much in

agreement with that postulated here for the time-scales of melting.

74: The authors are again confusing different terms. How does melting facilitate exhumation (a generic term describing the return of once deep-seated rocks to the Earth's surface)? It is not clear how weak rocks can be driven toward the surface if they are weak? Are the authors implying that the main rock exhumation mechanism in the Himalaya is diapirism? Maybe the authors really mean extrusion? This is an entirely different process that may, or may not, be related to exhumation.

76: The text is vague "long-lived soft regions lubricating the Himalayan orogen..." I have no idea what this means...

77: I don't understand this statement. How does the crust being weak sustain high elevations? Earlier, the authors argue that the crust is weak and therefore flows. But now, the authors argue that weak crust can sustain high elevations? Wouldn't it be the exact opposite? If the crust is weak high topography should not be sustained. There is little logic here.

92: "inactive nowadays" This language, here and throughout, is too imprecise. The reader has no idea what this might mean.

92 – 94: This sentence makes no sense. First part – "melting occurred at 10 – 12km depth", second part of same sentence "melting occurring at a depth of >20km".

So, which is it?

95: Again authors are confused between exhumation and extrusion. These are different processes, governed by different boundary conditions.

95: So "southward ductile exhumation" is occurring today? Not sure I know of any modern paper that has argued this.

190: mixture of pressure and depth reported which is confusing.

191: What does "depending on the amount of excess water" actually mean?

Supplemental Fig. 8: How do these rock samples relate to the paper? I cant find any information. However: 1) It is not possible to extract reliable PT information from these bulk compositions (they lack garnet as far as I can tell); 2) these are not the bulk compositions that produce large volumes of melt in the Himalaya (they are too quartzofeldspathic).

Reviewer #3 (Remarks to the Author):

Reviewed for Nature Communications by R.A. Jamieson, 2 October, 2017

General Comments: This is an interesting paper that tackles the long-standing question of whether resistivity anomalies beneath the Himalaya and Tibet represent fluid or melt, which in turn has implications for the tectonic evolution of the orogen. The work is novel in that it combines data from melting experiments with magneto-telluric data from the Himalaya in order to deduce melting conditions and fluid contents associated with observed MT anomalies. Fluid and melt proportions for specific bulk compositions at various P-T conditions are calculated from thermodynamic models. The results are compared with a PTt path taken from a previously published geodynamic

model to test hypotheses concerning the influence of melting on tectonics. If the approach has the predictive capacity claimed by the authors, this work will certainly interest a broad spectrum of earth scientists.

My main concern is with scaling – do the experimental data really apply quantitatively to the crustal scale? If the authors can address this and a few other questions (below), their claim to have developed a robust method to determine whether MT anomalies represent melt or fluid would be strengthened, and the paper should meet the criteria for publication (e.g., an advance in understanding likely to influence thinking in the field).

Detailed Comments (line #):

76 ff., Fig.1. It is difficult to relate the various points shown on the summary cross-sections (Fig. 1b,c) to the original MT data in the cited papers. The original profiles are in some cases not clearly identified (e.g. which Unsworth et al 2005 profile is used in 1c?), nor is it clear how points P1-P5 correspond to the MT anomalies (generally zones rather than points) in the original profiles. It would be very helpful if the original MT profiles could be reproduced somewhere, perhaps in the supplementary data, with the points P1-P5 superimposed on the corresponding anomalies.

125-142, Eqn 1-4. Both the melting experiments (new) and MT data (previously published) appear to be technically sound. My main concern is scaling: do conductivity measurements on mm-scale capsules scale appropriately to km-scale crust? In particular, can measurements on homogeneous glass charges reliably predict conductivity in the crust, where conductive materials (whether melt, fluid, or something else) must be heterogeneously distributed in fractures, shear zones, and intergranular networks? Have the authors tried to apply their method to a “representative volume”, as is standard practice, for example, in reservoir engineering? While it is obvious that relative comparisons can be made, the paper claims to make quantitative predictions of temperature-depth-conductivity-melt/fluid content for specific MT anomalies (P1-P5) beneath the Himalaya. These anomalies occupy different structural positions in the orogen and therefore differences in distribution/connectivity etc. of conductive materials might be expected. I did not see any mention of this issue in either the paper or the supplementary material. Because the predictions underlie the claim that the authors have developed a new and robust method for distinguishing melt from fluid, some additional supporting evidence or discussion is warranted.

153. A quick Google search revealed that there are > 1000 hot springs in Tibet, many of them popular tourist destinations. While scientific descriptions may be rare, the authors should not be too quick to discount fluid as a possible source of MT anomalies beneath the plateau.

176-177, Fig. 3. PTt path G1 does not appear in Beaumont et al. (2001) as implied here; it was first published by Jamieson et al 2004 (also cited, but not where G1 is first mentioned). The PT results presented by Beaumont et al. (2001) were from a different model, and did not include path G1. Clarify. In addition, path G1 (Fig. 3 inset) appears to be correct but is shown in the petrological (P increasing up) reference frame. For a paper making explicit comparisons with geophysical profiles, it would be more appropriate to use the geophysical (P increasing down) reference frame, as originally published by Jamieson et al. (2004). This would also be consistent with the P axis in Fig. 3b.

179-180, 260-261, Supp. Text 3, Supp. Table 4: How was T estimated for MT anomalies P1-P5? For the method to offer a robust prediction of melt vs fluid associated with present-day MT anomalies, an independent estimate of present-day T should be used in the calculations. It is not clear whether/how this was done, or if T was estimated from the melting experiments or by assuming the conditions of model path G1 (e.g., line 261). An explicit comment on this should be included in the main text.

248 ff, Fig. 4. Path G1 reflects the evolution of the model orogen over > 50 million years. The thermal state of the orogen evolved over that time, and continues to evolve today – i.e., the

system is not steady-state. It is therefore not clear that PT conditions inferred from path G1 at some early stage of orogenic evolution would apply quantitatively to specific parts of the system today. Further justification is warranted.

Editorial (line #):

The writing is OK, but there are quite a few relatively minor editorial problems that should be fixed before publication. I have not made detailed comments but a few examples follow:

35 and elsewhere – ...data are... (“data” is a plural word)

38-40: Major crustal melting means that material beneath the Tibetan plateau must behave as low viscosity regions..... (“melting” is a process and cannot “behave” as a region). Similar word-use and subject-verb problems can be found throughout the paper and in places affect the clarity.

52. ...associated with ... (not “associated to”)

53. ...solidified products.... (“fossils” is not an appropriate word in this context)

88. ...both currently inactive... (“nowadays” not appropriate in a formal paper) When were these structures active?

89.witness to.... (not “witness of”)

Point-by-point response to referee's comments

[Our responses] We sincerely thank all three reviewers for the insightful comments. We give our responses to the comments below. We have fixed the syntax errors, revised the main text and added some explanations in the manuscript as indicated below. In the revised ms, the changes are highlighted in red.

Reviewer #1 (Remarks to the Author):

Review of "Long-lived weak regions within the Himalayan crust due to water-rich granitic melts"

Reviewed by Susan Ellis, September 18th 2017

Overall impression: In general I liked this paper and thought it was a useful contribution to the debate about melt vs. fluids in Himalayan tectonics.

My main thought is that the paper did not "fess up" quite enough about a primary assumption that it makes- and which is critical to support the paper's conclusions. This is the assumption of a kind of "quasi steady-state" (mentioned right at the end of the paper on line 316). That is- that the present-day geophysical anomalies- particularly resistivity- can be related to the rock record from exhumed leucogranites. This assumption is made despite the fact that the main faults bounding the exhumed section (the MCT and STD) are currently inactive. I think the authors need to be a lot clearer about this assumption and to justify it more.

[Our responses] Thank you for this insightful comment. We have taken into account this point. Our conclusion is that while the thermal regime prevailing 20 Ma must still operate beneath the Tibetan plateau, the global tectonic regime and the prevailing faults have switched. Implicitly, there is therefore "lag" or a "decoupling" between the large scale tectonic and the thermal state of the crust. Before addressing the next point, we underline that we never hid the information that the STD has ceased in our ms.

They also need to spend more time in the introduction detailing the bounding parameters for activity on the thrust faults (timing vs. tectonics; role in exhuming the partial melt; when did activity cease on these faults?). Provided this is clarified I think the paper will make a nice addition to the literature.

[Our responses] We addressed that points in the introduction see paragraph "The Himalayan crust: field observations and geophysical signals".

Some minor comments:

1. fluid-absent vs. fluid-enabled terminology and explanations: I got rather confused by the loose way these terms were bandied around in the paper. I think the authors need to clearly define, near the beginning, what these terms mean- for resistivity vs. melting. For example, does fluid-enabled include cases where hydrous minerals are present- but no free water in pore spaces? How do these terms relate to the term w (wt%), and the term "water concentration (wt%" used in Figure S6)? And is w the same as label "H₂O in rocks" on figure S7?

[Our responses] The terminologies of fluid-absent melting vs. fluid-present (or fluid-enabled in your questions) melting were explained in the introduction but we have added important clarifications; the vocabulary used was addressing the petrological community; we now have modified our phrasing in order to make it accessible to the wide audience of *Nature Communications*. In particular, we make clear that water is always present during crustal melting

but the difference between fluid-absent and fluid-present melting is if free water exists or not before melting starts.

The water concentration (w) both in eq. (4) and Fig. S6 means the water content in the magmas; while "H₂O in bulk rocks (wt%)" in Fig. S7 (now Fig.4) refers to the H₂O in the whole rock undergoing partial melting (*i.e.* solid+liquid+fluid). In other words, "H₂O in bulk rocks (wt%)" includes both H₂O in the magmas (w) and free fluid phase which cannot be dissolved in the magmas because of fluid saturation. You can see in Fig.4, how the electrical conductivity is defined in the bulk "H₂O in the rock-temperature-melt fraction" parameter space: Here the parameter space is bounded by the fluid-absent melting ("H₂O in bulk rocks" equals to 1.74 wt%) and the fluid saturated regime (water is present in excess to the melt H₂O-solubility). In between, the regime of fluid-present melting operate.

What particularly confused me was the sentence on line 213, which claimed that "these melts" (referring to the P2 anomaly, which is claimed to have "fluid-absent melting along the G1 P-T path (line 209)) contain 6.5-7.1% wt% H₂O. Is this a typo? Surely these are then not fluid-absent?

[Our responses] We are sorry to say that it is not a typo. 6.5-7.1% wt% H₂O is the water concentration (w) of the magmas in the P2 position. The confusion is introduced by the term "fluid-absent", which does not mean at all dry. We hope this is now clear in the ms.

I was also confused by lines 222-223 where the % is separated into hydrous mineral water and additional fluid. And how does this relate to the "Adjusted H₂O" also mentioned in the manuscript. In figure caption 3, it says "(*i.e.* excess water content of ca. 1.2%)"- where did this come from? Is this free water?

[Our responses] These are two different conceptions:

"Adjusted H₂O" (see table S3) in the methods identifies the water concentrations (w) of our experimental samples (*i.e.* hydrated leucogranitic melts) measured by FTIR in order to fit the P-T-H₂O model (eq. (1)). "Adjusted H₂O" is forced to be between the water content of samples measured before and after the conductivity experiments (table 2).

The "1.2 wt% of H₂O" indeed indicated the excess free water phase in the metapelites before melting starts (*i.e.* fluid-present melting). Such excess fluids is consistent with the electrical conductivity beneath Tibet and the geochemical observations in some Himalayan leucogranites.

2. Figure 1 labels "present day"- not sure what this means as it is not explained in the figure caption

[Our responses] Thanks for pointing this mistake. The "present-day" was deleted as it adds some confusions and is not required at this stage of the paper..

3. Figure 3 does not plot P4 and P5 on the P-T paths

[Our responses] Indeed P4-P5 these correspond to pluton emplacement rather than partial melting. We now show P4-P5 in Fig. 5 (former Fig. 3) to show that it is clearly something different. We also built Fig. 6 to help visualizing the different P1-to-P5 spots in a scheme illustrating the magmatic processes from the source (melting) to the pluton emplacement (solidification).

4. Line 287. Please tell us what the extrusion rates are for comparison

[Our responses] There was a phrasing issue as extrusion is not the appropriate wording. We used the denaturation rates of the GHC assumed by Beaumont *et al.*¹²

5. line 299. "Leucogranite plutons...for such dykes". Are the two parts of the sentence referring to the same thing- are they plutons or dykes?

[Our responses] Dykes feed the plutons (rephrased see lines 454-455).

6. line 315-316. This sentence seems self-contradictory- see main comment above

[Our responses] We hope that this paragraph is now clearer.

7. English grammar- in some places it needs some work eg

"bear witness of" line 93- to rather than of?

line 172 "very well" sounds clumsy and not needed

lines 208-235. I found this whole section very confusing and at the end of it I was even more confused which anomalies are supposed to be fluid-enabled vs. dry. I suggest a careful rewrite including wt% thresholds used to define the terms and mineral bound vs. pore fluid mentioned.

lines 258-259- meaning not clear, rewrite

lines 266-267- ditto

line 269 "that fluids" perhaps say "that significant fluids"

Methods section

lines 570, 572 "On one side...on the other side" correct the grammar

line 574: "On the contrary"-> "In contrast"

line 597: "were realized" -> "were performed"

line 602: here is yet another definition of fluid-absent vs. fluid-present. Perhaps the confusion arises because the effect of fluids on melt vs. resistivity involved different thresholds? If so clarify this somewhere.

line 632: "with the parameter C and n"-> "Where parameters C and n"

[Our responses] Thanks for pointing out us these grammatical mistakes. We have checked and revised all these errors.

Reviewer #2 (Remarks to the Author):

Overall, I like the idea that the authors are advancing. Linking ancient rock record to modern geophysical data through experimental constraints. This has great potential to allow us to link melt process to product in a more quantitative manner.

However, in my view, the paper falls well short of the impact required for Nature. There are numerous issues with the paper that prevent me giving it my support. Below I briefly summarize my concerns.

Figure 4, in my mind, encapsulates the author's fundamental misunderstanding about the petrology and geochemistry of the granites exposed in the Himalaya, and how, if at all, they might relate to modern day geophysical data. The authors imply, based on the way that draw their figure, that there should be two types of granites exposed in the GHC (fluid-present and fluid-absent). The geochemistry of these rocks is distinct, and is easy to separate out. However, the fundamental problem is that inside the main exposure of the GHC, there are no rocks that were formed by fluid-present melting reactions. No single paper has reported such rocks. Sure, such rocks exist to the north of the main GHC, in domes, and further north (and some way to the south), but none are inside the GHC, as drawn in the cross-section. I raise this point, because it

is fundamental to the authors assertions about how melts form and how the Himalaya actually work.

[Our responses] Thank you for this comment. We realized that we indeed focused far too much our paper on the fluid-present melting that is deduced from the analysis of the P3 spots only, while we concluded that fluid-absent melting prevails in all other analyzed settings (see table 3). Our text has been accordingly rephrased. We agree also with the reviewer that fig 7 (former fig 4) was exaggerating the importance of fluid-absent and this has been modified.

Our study revealed that fluid-present melting is required to explain the P3 anomaly, yet, only 1wt. % of excess water is required. Is there a way to categorically reject this scenario? We do not think so and we believe that the reviewer will agree. The P3 anomaly is in indeed localized in the GHC in the geological section (Fig. 1c), but it is an interpretation; It may well be in a geological unit similar to that sourcing the THL granites outcropping further south (Fig.1). We also recall the geochemical survey by Guo & Wilson (ref. 31) (that we, however, do not endorse) arguing that most Himalayan granites (HHL or THL) derived from fluid-present melting, indicating that the role crustal fluids in the genesis of the Himalayan granite remains debated. We also just noted the recent paper by Harrisson (*American Mineralogist*, Volume 101, pages 1348–1359, 2016) reporting the occurrence of fluid-present melting for a pluton of the eastern-Himalaya.

In the text, we refer to the experimental survey by Patino-Douce & Harris (ref. 2), which may represent one of the rare quantitative assessment of how fluid-present melting may be detected: high Ca content in the melt due to plagioclase destabilization. However, with 1 % of excess-water, the changes in the melt composition is their experiments remains unnoticeable and the melt CaO is similar to that of leucogranites. All in all, we suggest that the highest electrical conductivity value requires high degree of melting (30%) producing water-rich melts and this requires 1% of excess water; This does not appear at odds with the literature of Himalayan granites, which as we show remains controversial.

In addition, Figure 4 shows a series of fluids entering the GHC from the subducting Indian plate. This may be feasible (setting aside the likely impermeability of the major bounding fault , HMT), but then, wouldn't fluid-present melts dominate the GHC? Why do such melts not dominate the geologic record inside the GHC, as exposed today?

[Our responses] We agree and decided to tone down the visibility of fluid-present melting in Fig. 7. But fluids are seen by MT ahead of the MCT as shown in Fig. 7 (former fig 4). We even wonder if melting does not occur also ahead of the MCT. Finally, nothing tell us that the "subduction" of water-rich lithologies operating today was playing a critical role in the partial melting of the GHC 20 Ma. We just say that the thermal state of the GHC does not seem to differ to that prevailing 20 Ma.

Finally, the authors assert that melts are able to cross the STDS. Time and again, such observations have either proven to be inaccurate (Hodges in Everest), or that they are in fact beneath the main STDS (Carosi in W.Nepal). The timing of movement, now very well constrained, does not permit melts to cross the STDS.

[Our responses] Thank you for pointing this out. We cited these papers and others on this topic (ref. 27-29). Maybe, we gave the feeling that this is an important step in our demonstration, but our work demonstrate that the depth and magnitude of the MT anomalies matches the dynamic of crustal melting prevailing 20 Ma; whether melt can go through the STD or not, is an important topic re the dynamic of melt transfer in the Himalaya, but probably not so central for our study. We however do not see why the STD being now inactive would play the role of an impermeable barrier it was believe to play 20 Ma when it was active. We underline that the THL (granite intruding the THS units, see Fig. 1) may derive from the GHC (*i.e.* Guo & Wilson³¹) implying that dykes may have gone through the STD. We also make clear in the ms that we recognize that

plutons are certainly not able to do so, but dykes may go through the detachment (e.g. Caloisi *et al.*²⁹ have observed dykes crossing the STD).

There is some major confusion with terminology and existing literature that results in the meaning of the ms. being lost in several key places.

Line 61: “granite-related migmatites and granulites”

The authors are confused. A granulite does not by its definition necessarily involve partial melting, or extraction of melt from a rock.

[Our responses] Sorry for this rough sentence that has been deleted. Thank you for pointing that out.

62: “but it is always difficult to distinguish the timing of individual events”

This is simply untrue. The authors ignore many ten’s of papers that report high precision, high resolution ages, and for granitic systems, particularly in the Himalaya. There is a vast body of knowledge of this topic that is entirely ignored here, and elsewhere in the literature. As but one example, Lederer *et al.*, 2013, CMP, report numerous high resolution isotope geochronology ages of granites from NW India. These authors were able to construct a model, that is very much in agreement with that postulated here for the time-scales of melting.

[Our responses] This is indeed an uneven phrasing that has been modified. This ref has been added (ref. 14).

74: The authors are again confusing different terms. How does melting facilitate exhumation (a generic term describing the return of once deep-seated rocks to the Earth's surface)? It is not clear how weak rocks can be driven toward the surface if they are weak? Are the authors implying that the main rock exhumation mechanism in the Himalaya is diapirism? Maybe the authors really mean extrusion? This is an entirely different process that may, or may not, be related to exhumation.

[Our responses] Thank you for raising this imperfection of phrasing. The sentences have been modified.

We indeed suggest that >>7% melt content in the middle crust of GHC must significantly weaken crustal rocks and the viscosity of the large regions impacted by such melting is of the order of 10^{11} Pa·s. So, such soft regions could lubricate the Himalayan Main Frontal Thrust resulting in the southward exhumation of the GHC. Diapirism is not what we suggested, but buoyancy due to vast episodes of melting may play a role (this hypothesis is not in the paper).

76: The text is vague “long-lived soft regions lubricating the Himalayan orogen...” I have no idea what this means...

[Our responses] The text has been modified and the vague sentence has been deleted (see around line 474).

77: I don’t understand this statement. How does the crust being weak sustain high elevations? Earlier, the authors argue that the crust is weak and therefore flows. But now, the authors argue that weak crust can sustain high elevations? Wouldn’t it be the exact opposite? If the crust is weak high topography should not be sustained. There is little logic here.

[Our responses] It is true that this sentence seems little supported by logic. This has been deleted.

92: “inactive nowadays” This language, here and throughout, is too imprecise. The reader has no idea what this might mean.

[Our responses] Corrected

92 – 94: This sentence makes no sense. First part – “melting occurred at 10 – 12km depth”, second part of same sentence “melting occurring at a depth of >20km”. So, which is it?

[Our responses] This has been rephrased: melting occurs at depth of >20 km, while plutons are emplaced at 10-12 km; fig 6 makes a clear and visual illustration of this process that has been difficult to capture by at least two of the reviewer. We hope this new figure will help.

95: Again authors are confused between exhumation and extrusion. These are different processes, governed by different boundary conditions.

[Our responses] Corrected, thank you for helping us with these language subtleties.

95: So “southward ductile exhumation” is occurring today? Not sure I know of any modern paper that has argued this.

[Our responses] We have deleted this.

190: mixture of pressure and depth reported which is confusing.

[Our responses] Thanks for pointing out us this confusion. We have changed the “ca. 35 km depth” to “1.22 GPa”.

191: What does “depending on the amount of excess water” actually mean?

[Our responses] This has been clarified in the text. Excess water is what is causing fluid-present melting. We made several calculation with “excess water” ranging from zero (fluid-absent melting) to ca. 3% excess water (see Fig. 4).

Supplemental Fig. 8: How do these rock samples relate to the paper? I cant find any information. However: 1) It is not possible to extract reliable PT information from these bulk compositions (they lack garnet as far as I can tell); 2) these are not the bulk compositions that produce large volumes of melt in the Himalaya (they are too quartofeldspathic).

[Our responses] This was indeed confusing and the text has been changed and simplified (see lines 607-610). We need an assessment of the grain size in the GHC to calculate permeability and melt segregation during melting. We considered the average grain size d of the anatectic metapelite in the GHC given by Groppo, *et al.*³³ and we performed a few assessment of the grain size of unmelted gneisses.

Reviewer #3 (Remarks to the Author):

Reviewed for Nature Communications by R.A. Jamieson, 2 October, 2017

General Comments: This is an interesting paper that tackles the long-standing question of whether resistivity anomalies beneath the Himalaya and Tibet represent fluid or melt, which in turn has implications for the tectonic evolution of the orogen. The work is novel in that it combines data from melting experiments with magneto-telluric data from the Himalaya in order to deduce melting conditions and fluid contents associated with observed MT anomalies. Fluid and

melt proportions for specific bulk compositions at various P-T conditions are calculated from thermodynamic models. The results are compared with a PTt path taken from a previously published geodynamic model to test hypotheses concerning the influence of melting on tectonics. If the approach has the predictive capacity claimed by the authors, this work will certainly interest a broad spectrum of earth scientists.

My main concern is with scaling – do the experimental data really apply quantitatively to the crustal scale? If the authors can address this and a few other questions (below), their claim to have developed a robust method to determine whether MT anomalies represent melt or fluid would be strengthened, and the paper should meet the criteria for publication (e.g., an advance in understanding likely to influence thinking in the field).

[Our responses] This is indeed the Holy Grail for experimentalist and geodynamicist with no definitive answer as the topic is vast. Yet, we wrote the paragraph lines 278-298 in order to address this scaling issue and the following paragraph (line 299-318) to clarify further the methodology of lab-to-field conversion. We discuss the frequency range (MT vs lab), and the scale (spatial resolution of MT vs field heterogeneity). Our lab data are parameterized (eq.1-4) and directly coupled with petrological models of partial melting and a physical law calculating the conductivity of a two-phase mixture. This strategy allows us to convert the main petrological parameters, that is to say, temperature and water content, into an electrical signal.

Detailed Comments (line #):

76 ff., Fig.1. It is difficult to relate the various points shown on the summary cross-sections (Fig. 1b,c) to the original MT data in the cited papers. The original profiles are in some cases not clearly identified (e.g. which Unsworth et al 2005 profile is used in 1c?), nor is it clear how points P1-P5 correspond to the MT anomalies (generally zones rather than points) in the original profiles. It would be very helpful if the original MT profiles could be reproduced somewhere, perhaps in the supplementary data, with the points P1-P5 superimposed on the corresponding anomalies.

[Our responses] Supplementary Fig. 1 has been provided as the reviewer demanded. It relates both studies of Unsworth *et al.*¹⁷ and Arora *et al.*³⁵ to the P1-to-P5 anomalies shown in Fig.1; the latter being an adaptation of Yin *et al.*²⁵

125-142, Eqn 1-4. Both the melting experiments (new) and MT data (previously published) appear to be technically sound. My main concern is scaling: do conductivity measurements on mm-scale capsules scale appropriately to km-scale crust? In particular, can measurements on homogeneous glass charges reliably predict conductivity in the crust, where conductive materials (whether melt, fluid, or something else) must be heterogeneously distributed in fractures, shear zones, and intergranular networks?

[Our responses] We use appropriate 2-phase laws as already mentioned in our 1st response to Prof. Jamieson. We did not consider possible heterogeneous distribution in the sense suggested by the reviewers because the MT data does not indicate such complexities. No electrical anisotropies has been suggested for example.

Have the authors tried to apply their method to a “representative volume”, as is standard practice, for example, in reservoir engineering? While it is obvious that relative comparisons can be made, the paper claims to make quantitative predictions of temperature-depth-conductivity-melt/fluid content for specific MT anomalies (P1-P5) beneath the Himalaya. These anomalies occupy different structural positions in the orogen and therefore differences in distribution/connectivity etc. of conductive materials might be expected. I did not see any mention of this issue in either the paper or the supplementary material. Because the predictions underlie the claim that the authors have developed a new and robust method for distinguishing melt from fluid, some additional supporting evidence or discussion is warranted.

[Our responses] We understand the comment. We believe that by addressing the above comments we have already addressed Prof Jamieson's comment. Implicitly, the concept of representative volume is given by the spatial resolution of the MT analysis at the depth of the anomalies (500 m, see text, Lines 278-298). As there is, however, no guarantee that the crust is homogenous at such a spatial scale, we assumed it. This is made clear in the text.

153. A quick Google search revealed that there are > 1000 hot springs in Tibet, many of them popular tourist destinations. While scientific descriptions may be rare, the authors should not be too quick to discount fluid as a possible source of MT anomalies beneath the plateau.

[Our responses] Thank you for pointing out this. Indeed, we have read several papers and cited (ref. 51-52) on this topic. We discovered that the geochemical features of these hot springs (those at the south-tibet) perfectly match a source similar to that of the leucogranites. Helium, carbon and nitrogen isotopes clearly speak for it. The water must result from rain fall infiltrations, heating and upwelling by advection, but surely, part of this water is expelled during granite solidifications at 10-12 km depth. We recall here that the volume of the MT anomalies represent a vast amount of water being cycled through the orogeny (see Fig 6-7).

176-177, Fig. 3. PTt path G1 does not appear in Beaumont et al. (2001) as implied here; it was first published by Jamieson et al 2004 (also cited, but not where G1 is first mentioned). The PT results presented by Beaumont et al. (2001) were from a different model, and did not include path G1. Clarify.

[Our responses] Thanks for this comment. We have revised the text accordingly.

In addition, path G1 (Fig. 3 inset) appears to be correct but is shown in the petrological (P increasing up) reference frame. For a paper making explicit comparisons with geophysical profiles, it would be more appropriate to use the geophysical (P increasing down) reference frame, as originally published by Jamieson et al. (2004). This would also be consistent with the P axis in Fig. 3b.

[Our responses] Thanks for suggesting us to adjust the inset diagram of Fig. 5 (former fig3a). We have modified this diagram and it looks more consistent now.

179-180, 260-261, Supp. Text 3, Supp. Table 4: How was T estimated for MT anomalies P1-P5? For the method to offer a robust prediction of melt vs fluid associated with present-day MT anomalies, an independent estimate of present-day T should be used in the calculations. It is not clear whether/how this was done, or if T was estimated from the melting experiments or by assuming the conditions of model path G1 (e.g., line 261). An explicit comment on this should be included in the main text.

[Our responses] We clarified the text but yes, T profiles assume the G1 path and it perfectly explains the MT and seismic observations. So, we consider this as a validation of our assumption.

248 ff, Fig. 4. Path G1 reflects the evolution of the model orogen over > 50 million years. The thermal state of the orogen evolved over that time, and continues to evolve today – i.e., the system is not steady-state. It is therefore not clear that PT conditions inferred from path G1 at some early stage of orogenic evolution would apply quantitatively to specific parts of the system today. Further justification is warranted.

[Our responses] We understand the reviewer's concerns, but this is the simplest explanation: the seismic and electrical properties of the present-day crust is perfectly explained by the persistence of the G1 P-T path. No other assumptions are required.

Editorial (line #):

The writing is OK, but there are quite a few relatively minor editorial problems that should be fixed before publication. I have not made detailed comments but a few examples follow:

35 and elsewhere – ...data are... (“data” is a plural word)

38-40: Major crustal melting means that material beneath the Tibetan plateau must behave as low viscosity regions..... (“melting” is a process and cannot “behave” as a region). Similar word-use and subject-verb problems can be found throughout the paper and in places affect the clarity.

52. ...associated with ... (not “associated to”)

53. ...solidified products.... (“fossils” is not an appropriate word in this context)

88. ...both currently inactive... (“nowadays” not appropriate in a formal paper) When were these structures active?

89.witness to.... (not “witness of”)

[Our responses] Thanks for helping us to correct these grammatical mistakes. We have revised the text accordingly.

END.

Reviewers' comments:

Reviewer #1 (Remarks to the Author):

The authors appear to have addressed my concerns adequately. It's good that they have spent some effort making the paper accessible by a wider audience.

sincerely
Susan Ellis

Reviewer #2 (Remarks to the Author):

I reviewed an earlier version of this manuscript. As with the previous iteration, I am supportive of the general concept of linking ancient and active melt systems to unravel the history of collisional orogenic belts. However, I feel that the paper still falls well short of the impact and high standard of Nature. I briefly outline some of the reasons below:

1. The abstract does not address the subject. For example, Line 29, the authors state in the abstract

"we address this problem by connecting pressure-temperature paths recorded in ancient exhumed crustal rocks from the Himalaya to the present-day conditions beneath the Tibetan plateau imaged with geophysical data".

There are two problems here: 1) The authors never do this, either directly or indirectly. I see no summary of any kind of PT constraints on crustal rocks summarized or compared to the geophysical data. 2) A simple comparison by the authors of such data would reveal that the geological data from the Himalaya do not match that of the modern day plateau. For example, the plateau is said to be at 1000C at lower crustal conditions (work by Hacker and colleagues), whereas no rocks in the GHS have ever been measured to be that hot.

2. Figure 5 contains a PT path inset from Groppo et al., This is a fairly specific PT path for one package of rocks and is not generally applicable across the orogen. This PT path predicts melting on the prograde path as well as retrograde path. There is a mysterious grey shaded region on the inset that is not described, but this is not the "melting field" (melting is not isobaric). Given the lack of real PT constraints in this paper, the comparison is fictitious.

3. As in my previous review, I maintain that the authors do not understand the geochemical difference between fluid-present, and fluid-absent melts, as recorded in the Greater Himalayan Series. For example, figure 6, indicates that only melts where fluid is in excess results in migration and formation of large plutonic bodies (plutons are only present on the left side of the figure). This figure is extremely confusing and I have a hard time figuring out exactly what is going on. For example, where in the Himalaya is there evidence of >8km of vertical continuous melt transport? Simple conduction models indicate cooling of granitic liquid over extremely short vertical distances – how do the authors explain 8km transport?

4. The new text contains new mistakes and does not match the geological data. For example: Line 490 "Our interpretation in Fig. 7 allows the melt to go through (via dykes and not via diapirs) the STD as long as the latter is inactive. This may conflict with the field analysis of plutons that emplaced 20 Ma closed to the STD, but this is the best explanation to the present-day P4 and P5 anomalies"

This makes no sense. The STDS is active until at least 13 Ma along the majority of the orogen. If the explanation conflicts with the field data, then how can the explanation be correct? The authors

show a lack of understanding of the basic geology of the Himalaya or of research into granites in the GHS, and work that has been done to understand their timing and style of emplacement relative to the STDS.

I could carry on with citing additional examples where the paper makes no sense, but those listed above serve as representative reasons that I cannot support publication of this paper because it ignores basic field observations and geochemical constraints and does not really solve the problem that is set out in the abstract.

Reviewer #3 (Remarks to the Author):

The paper has addressed some of the issues raised by previous reviewers. Some problems remain, as follows (line numbers refer to revised version):

1. (287-90) Scaling: The issue of sample size vs conductivity has been addressed. However, the question of homogeneity and connectivity remains. GHS migmatites are highly heterogeneous on the outcrop scale. Can they be assumed to be homogeneous (or at least made up of interconnected melt networks) on the scale of typical MT sampling volumes? This is hinted at (lines 305, 366, 398) but should be addressed explicitly. In addition, how much melt is present at any one time in a system like this?
2. (295-96) Have 30% uncertainties in MT data been taken into account in interpreting the melt/fluid contents associated with anomalies P1-P5?
3. (298-355) Assumed steady-state and applicability of path G1 to present-day study areas: Model results from HT1 (Jamieson et al. 2004) show how the model thermal structure has evolved over the last 25 million years – definitely not steady-state beneath the orogenic front, although the model thermal structure beneath the Tibetan plateau has been more stable. Path G1 tracks a specific particle through the model orogen from 54 Ma to the present; the P-T conditions “seen” by this particle do not necessarily apply to other particles in other places and at other times. While the results are interesting the authors should be careful how they interpret the melting model for G1 in terms of present-day conditions in different parts of the system. A more fruitful or complementary approach might be to compare the MT profiles with model isotherms from HT1 at 0 Ma at equivalent depths and distances from the orogenic front. For example, Jamieson & Beaumont (GSAB 2013) compared model viscosity with MT data (their Fig. 16B, attached). The viscosity contour in this diagram (10^{19} Pa.s ~ onset of melt-weakening) lies between the 700 & 800 C isotherms. The comparison supports the interpretation that the MT anomalies represent melt. We did not attempt to link model results to % melt or % water in melt, but the authors could use the 0 Ma model temperatures as a constraint on linking their experimental data to the MT measurements.
4. (415-425) If LHS is the source of the fluid, it should lie below the region of active melting, which does not appear to be the case. Present-day LHS rocks were accreted to the footwall of the overriding GHS during extrusion (prograde metamorphism happened after melt generation) and cannot have been the direct source of fluid for fluid-present melting.
5. (450 ff) Tectonic interpretation: The final section of the paper has been greatly expanded from the original version, possibly in response to comments from reviewer 2. However, most of this expanded section (especially after ca line 450) is highly speculative and in many places reveals confusion and/or inconsistency with respect to tectonic interpretations. In my opinion, this section weakens the impact of the paper and should be removed. The authors should focus on strengthening their melt vs magma interpretation, which is an important contribution on its own. The tectonic speculation may be a logical extension of this work but it belongs elsewhere.

The writing still requires quite a lot of tidying up. I have attached an annotated pdf with a number of scientific comments and editorial suggestions.

R.A. Jamieson
23 February 2018

REBUTTAL LETTER FOR NCOMMS-17-20777A

Reviewer #2 (Remarks to the Author):

I reviewed an earlier version of this manuscript. As with the previous iteration, I am supportive of the general concept of linking ancient and active melt systems to unravel the history of collisional orogenic belts. However, I feel that the paper still falls well short of the impact and high standard of Nature. I briefly outline some of the reasons below:

We understand the position of the reviewer and we respect his point of view. The ms we are re-submitting has been significantly modified and improved. The interpretation has been developed since we conclude that North-Western Tibet must be undergoing partial melting under conditions similar to those prevailing during the Miocene (ie. fluid-absent and P-T conditions similar to G1), while Southern Tibet have been modified since the Miocene. We then suggest that either the Southern Tibet crust is $\sim 100^{\circ}\text{C}$ warmer than it was during the Miocene or, 1% of H₂O has been introduced in the crustal lithologies increasing the degree of partial melting; both effects (increased temperature and H₂O content) can explain the higher conductivity.

1. The abstract does not address the subject. For example, Line 29, the authors state in the abstract

“we address this problem by connecting pressure-temperature paths recorded in ancient exhumed crustal rocks from the Himalaya to the present-day conditions beneath the Tibetan plateau imaged with geophysical data”.

There are two problems here: 1) The authors never do this, either directly or indirectly. I see no summary of any kind of PT constraints on crustal rocks summarized or compared to the geophysical data. 2) A simple comparison by the authors of such data would reveal that the geological data from the Himalaya do not match that of the modern day plateau. For example, the plateau is said to be at 1000C at lower crustal conditions (work by Hacker and colleagues), whereas no rocks in the GHS have ever been measured to be that hot.

The abstract has been accordingly modified. We do not fully understand this comment, despite several discussions among the co-authors. The G1 P-T path is in good agreement with thermobarometric studies on rocks from the GHS and we therefore decided to use it as a reference rather than proposing more specific and local P-T paths. Perhaps, the reviewer was suggesting that there are other P-T paths which are distinct from the G1 P-T path and which do not agree with thermobarometric studies on GHS rocks? In that case, the reviewer's comments should be addressed to the authors of the G1 model (Jamieson et al., 2004, ref. 18). Our opinion is that the G1 P-T path should be used as the most general P-T path – a point reinforced because it matches the MT model of the North-Western Himalaya extremely well. For Southern Tibet, we find that hotter temperatures are required or 1% H₂O must be added to the system to match the conductivity. These are a conclusions based on a quantitative assessment which also consider the uncertainty inherent in the calculations (list Table 3).

Re Hacker et al, we note that this is not a paper on the thermal state of the Tibetan crust and is not directly applicable to our study area in the Himalaya. We believe that the reviewer is mentioning the thermal state of the base of the crust being northward of the zone of interest for our paper, where mafic (and hot) magma have been found. The region discussed in our paper is Southern Tibet, and there, mafic magma is not expected (eg. He isotopes in granite and hot-springs).

2. Figure 5 contains a PT path inset from Groppo et al., This is a fairly specific PT path for one package of rocks and is not generally applicable across the orogen. This PT path predicts melting on the prograde path as well as retrograde path. There is a mysterious grey shaded region on the inset that is not described, but this is not the “melting field” (melting is not isobaric). Given the lack of real PT constraints in this paper, the comparison is fictitious.

More precisely, figure 5 shows the G1 P-T path, which is considered as an average broadly representative of the GHS rocks; indeed, this G1 P-T path agrees with Groppo et al (2012) and others as we cited in the paper. We agree with the questions about how representative path G1 can be considered, and have added a few sentences to explain that the G1 P-T path is an average P-T path for the orogeny. It certainly cannot account for all regional variations reported by Himalayan field studies (see line 363-370). The grey shaded area is a remnant symbol of a previous version that we forgot to remove – thank you for pointing this out.

3. As in my previous review, I maintain that the authors do not understand the geochemical difference between fluid-present, and fluid-absent melts, as recorded in the Greater Himalayan Series. For example, figure 6, indicates that only melts where fluid is in excess results in migration and formation of large plutonic bodies (plutons are only present on the left side of the figure). This figure is extremely confusing and I have a hard time figuring out exactly what is going on. For example, where in the Himalaya is there evidence of >8km of vertical continuous melt transport? Simple conduction models indicate cooling of granitic liquid over extremely short vertical distances – how do the authors explain 8km transport?

Fig 6 has been changed; there was indeed an unfortunate error in the drawing that none of us noticed. We thank the reviewer for pointing this out.

We realise that some of the reviewers concerns may be due to the phrasing of our paper. We have clarified this on lines 292-297. By fluid-present melting, we mean the melting of a rock having 2.9 wt% H₂O bulk. This means that at the onset of partial melting, the system is indeed fluid-present, but as a few percent of liquid forms, all the water is incorporated in the melt, making the system fluid-absent. We used the term fluid-present as the alternative to dehydration melting (i.e. 2.9 vs. 1.7 wt% H₂O bulk) as many specialists do (see ref. 6-9, 24). Our feeling is that the reviewer understands that we assume the presence of fluid all along the P-T path of melting. This is not the case and this is why we asked in our previous rebuttal letter for the reviewer to justify that melting a rock with 2.9 % H₂O instead of 1.7% can be excluded. The response to this question is NO as the experimental constrains on fluid present melting with 1% excess water do not produce results that significantly differs from fluid-absent as we underline in our previous rebuttal letter. This being said, and we hope the reviewer will appreciate this point: we propose the alternative that temperatures 100°C hotter than the G1 P-T path would also fit the MT observations in the Southern Tibet.

The comment about melt transport is somewhat confusing. The granitic melt was produced by partial melting at depths >20 km while the plutons are emplaced at 10-12 km. In between, dykes have been the conduit to transport granitic melts to plutons. We would refer the reviewer to references 24 & 32 in the ms. We cannot see why this is a problem.

4. The new text contains new mistakes and does not match the geological data. For example: Line 490 "Our interpretation in Fig. 7 allows the melt to go through (via dykes and not via diapirs) the STD as long as the latter is inactive. This may conflict with the field analysis of plutons that emplaced 20 Ma closed to the STD, but this is the best explanation to the present-day P4 and P5 anomalies"

This makes no sense. The STDS is active until at least 13 Ma along the majority of the orogen. If the explanation conflicts with the field data, then how can the explanation be correct? The authors show a lack of understanding of the basic geology of the Himalaya or of research into granites in the GHS, and work that has been done to understand their timing and style of emplacement relative to the STDS.

I could carry on with citing additional examples where the paper makes no sense, but those listed above serve as representative reasons that I cannot support publication of this paper because it ignores basic field observations and geochemical constraints and does not really solve the problem that is set out in the abstract.

We agree on this point, and these detailed examples will result in endless discussions that are not central for the paper. That section has been removed. Notice that reviewer 3 also demanded this part to be removed.

Overall some important changes have been made to the ms. We believe that the technical issues that were raised by the reviewer have been resolved and we thank the reviewer for some insightful comments.

Reviewer #3 (Remarks to the Author):

The paper has addressed some of the issues raised by previous reviewers. Some problems remain, as follows (line numbers refer to revised version):

1. (287-90) Scaling: The issue of sample size vs conductivity has been addressed. However, the question of homogeneity and connectivity remains. GHS migmatites are highly heterogeneous on the outcrop scale. Can they be assumed to be homogeneous (or at least made up of interconnected melt networks) on the scale of typical MT sampling volumes? This is hinted at (lines 305, 366, 398) but should be addressed explicitly. In addition, how much melt is present at any one time in a system like this?

To address this point, a series of paragraphs have been added to the ms in the section entitled: Laboratory to crustal scale conductivities (line 280-358). We hope that what the reviewer will appreciate this revision. We also hope that the reviewer appreciates that it is virtually impossible to address this point since it is very difficult to measure the inhomogeneity of migmatites over such a range of spatial scales. The MT data that are discussed in the ms would not allow identification of bodies having length scales of the order of 500 meters as observed in outcrops. Rather the MT gives a spatial average and reveals electrical resistivity anomalies having sizes of about 10^*10^*10 km; this could hardly be connected to the level of homogeneity/heterogeneity that any meter-scale outcrops indicate.

2. (295-96) Have 30% uncertainties in MT data been taken into account in interpreting the melt/fluid contents associated with anomalies P1-P5?

Yes (see fig 5), and we added an estimation of how such uncertainties propagate in terms of temperature, melt content and melt water content in Table 3.

3. (298-355) Assumed steady-state and applicability of path G1 to present-day study areas: Model results from HT1 (Jamieson et al. 2004) show how the model thermal structure has evolved over the last 25 million years – definitely not steady-state beneath the orogenic front, although the model thermal structure beneath the Tibetan plateau has been more stable. Path G1 tracks a specific particle through the model orogen from 54 Ma to the present; the P-T conditions “seen” by this particle do not necessarily apply to other particles in other places and at other times. While the results are interesting the authors should be careful how they interpret the melting model for G1 in terms of present-day conditions in different parts of the system. A more fruitful or complementary approach might be to compare the MT profiles with model isotherms from HT1 at 0 Ma at equivalent depths and distances from the orogenic front. For example, Jamieson & Beaumont (GSAB 2013) compared model viscosity with MT data (their Fig. 16B, attached). The viscosity contour in this diagram (10^{19} Pa.s ~ onset of melt-weakening) lies between the 700 & 800 C isotherms. The comparison supports the interpretation that the MT anomalies represent melt. We did not attempt to link model results to % melt or % water in melt, but the authors could use the 0 Ma model temperatures as a constraint on linking their experimental data to the MT measurements.

Thank you for this comment. A ref has been added was added to Jamieson & Beaumont (2013, ref. 60). A discussion has been added explaining the reasons why we have continued to use the P-T path G1 as a reference (368-375, see also 286-292). This is because this P-T path is a broad and average temperature profile deduced from modelling at a scale relatively similar to that of the MT observations, but also because the G1 P-T path has been corroborated by some field data. The zero age in Jamieson & Beaumont (2013) has not been corroborated by any field observations. It is an interesting prediction, but not an observation. The fact that the thermal status of P-T path G1 matches the melting condition during the Miocene and the present-day MT observations is a rather robust indication that the Jamieson and Beaumont (2013) zero-age prediction is not that correct for the North-Western Himalaya. In contrast, re Southern Tibet, a crust with a temperature 100°C higher than G1 at the Miocene, is quite possible given the prediction of Jamieson and Beaumont (2013) at zero age. We believe that the next generation of geodynamic models should link the Jamieson and Beaumont (2013) prediction to the MT observations using the methodologies that are developed in this paper. This would actually link the thermal state to the MT observation and allow some of the

thermomechanical model predictions to be validated or rejected. This is beyond the scope of the current paper and should be the subject of a future paper

4. (415-425) If LHS is the source of the fluid, it should lie below the region of active melting, which does not appear to be the case. Present-day LHS rocks were accreted to the footwall of the overriding GHS during extrusion (prograde metamorphism happened after melt generation) and cannot have been the direct source of fluid for fluid-present melting.

This section has been corrected and is now in agreement with Figure 7 and with the geometry of the Himalayan chain.

5. (450 ff) Tectonic interpretation: The final section of the paper has been greatly expanded from the original version, possibly in response to comments from reviewer 2. However, most of this expanded section (especially after ca line 450) is highly speculative and in many places reveals confusion and/or inconsistency with respect to tectonic interpretations. In my opinion, this section weakens the impact of the paper and should be removed. The authors should focus on strengthening their melt vs magma interpretation, which is an important contribution on its own. The tectonic speculation may be a logical extension of this work but it belongs elsewhere.

All this section has been removed.

The writing still requires quite a lot of tidying up. I have attached an annotated pdf with a number of scientific comments and editorial suggestions.

We have incorporated all the remarks of the annotated ms. Thank you for this effort.

We close this response with a remark that is addressed to all reviewers. All the conclusions and the approach deployed in our ms would not be possible without the high-quality unique data only produced in our lab ie. the conductivity of hydrous melts using a 4-electrode setup at variable P-T. We regret that none of the reviewers have appreciated this fact. We admit that this may have been because some of the discussion about the connection of lab and crustal measurements was poorly worded. We thank the reviewers for their comments which have improved the paper in this regard.

Best Regards,

The authors.

REVIEWERS' COMMENTS:

Reviewer #3 (Remarks to the Author):

As requested, I have looked over the new response to the referees (#2 & 3). Based on the rebuttal, I believe the authors have addressed the concerns adequately. In my opinion, the paper can now be published without further revision.